# Soil structure is an important omission in Earth System Models

Simone Fatichi [1*], Dani Or [2,3], Robert Walko[4], Harry Vereecken[5], Michael H. Young [6], Teamrat A. Ghezzehei [7], Tomislav Hengl [8], Stefan Kollet[5], Nurit Agam [9] & Roni Avissar[4]

Most soil hydraulic information used in Earth System Models (ESMs) is derived from pedo-transfer functions that use easy-to-measure soil attributes to estimate hydraulic parameters. This parameterization relies heavily on soil texture, but overlooks the critical role of soil structure originated by soil biophysical activity. Soil structure omission is pervasive also in sampling and measurement methods used to train pedotransfer functions. Here we show how systematic inclusion of salient soil structural features of biophysical origin affect local and global hydrologic and climatic responses. Locally, including soil structure in models significantly alters infiltration-runoff partitioning and recharge in wet and vegetated regions. Globally, the coarse spatial resolution of ESMs and their inability to simulate intense and short rainfall events mask effects of soil structure on surface fluxes and climate. Results suggest that although soil structure affects local hydrologic response, its implications on global-scale climate remains elusive in current ESMs.

[1] Institute of Environmental Engineering, ETH Zurich, Zurich, Switzerland. [2] Department of Environmental Science, Institute of Biogeochemistry and Pollutant Dynamics, ETH Zurich, Zurich, Switzerland. [3] Desert Research Institute, Reno, Nevada, USA. [4] Rosenstiel School of Marine and Atmospheric Science, University of Miami, Miami, Florida, USA. [5] Agrosphere, Jülich Research Center, Kreis Düren, Rheinland, Germany. [6] Bureau of Economic Geology, The University of Texas at Austin, Austin, Texas, USA. [7] Life and Environmental Sciences, University of California, Merced, Merced, California, USA. [8] OpenGeoHub foundation, Wageningen, The Netherlands. [9] Blaustein Institutes for Desert Research, Ben Gurion University of the Negev, Beersheba, Israel. *email: simone.fatichi@ifu.baug.ethz.ch

Soil is the domain where atmospheric and hydrologic processes are linked with the biosphere and biogeochemical cycles, thus playing a central role in supporting Earth's life[1–3]. About 40% of terrestrial precipitation returns to the atmosphere through the soil–plant–atmosphere continuum[4,5]. Nearly all the terrestrial annual global vegetation production and associated nutrient cycles rely on soil processes[6,7]. Soil properties, vegetation attributes, and land-use patterns jointly shape surface energy and water fluxes and regional climate[8–13], and affect extreme events such as heat waves and droughts[14–16]. More specifically, soil moisture and texture have been shown to affect weather[17,18], surface evaporation[19,20], and temperature extremes[21,22]. Soil hydraulic properties control soil water fluxes downward toward the groundwater table or laterally to stream networks, thus affecting groundwater and surface water resources[23–25]. The representation of soil processes at profile or plot scales, originally aimed at describing local phenomena (e.g., water balance of a field), has been extended to regional and global scales of Earth System Models (ESMs) with limited and uncertain upscaling considerations[26]. One of the challenges has been the systematic representation of the effects of soil structure. Even at the profile scale, quantifying the role of soil structure on infiltration, soil water fluxes, and transport processes remains a challenge[27,28], and is often neglected in many vadose zone studies. It comes as no surprise that parameterization of soil hydraulic properties for ESMs relies heavily on easy-to-measure soil textural maps with practically no consideration of soil structure and pedogenic information. This apparent omission is implicit in pedotransfer functions (PTFs) that are used for deducing soil hydraulic functions, such as water retention or hydraulic conductivity (e.g., see refs. [29,30]) from (primarily) soil textural information[31]. Consequently, despite the significant and well-established influence of soil structure on soil hydraulic functions (e.g., see refs. [32–36]) and its impacts on soil ecology (e.g., see refs. [37,38]), only a few studies have attempted to incorporate soil structure into PTFs[39,40].

To bridge this gap and assess its significance for the present hydro-climatic modeling, we propose a simple parameterization to include soil structure of biophysical origin (ignoring abiotic structural features) on local and global water fluxes and climatic attributes. Soil structure affects primarily the soil hydraulic conductivity function and, to lesser extent, the water retention curve. These modifications are approximated and are used to assess impacts on subsurface and surface fluxes at local and global scales. We tested the modified soil parameterization on eco-hydrologic responses in 20 representative locations worldwide using an ecosystem model (Tethys-Chloris (T&C), see Methods). In addition, we used a global climate model (Ocean-Land-Atmosphere Model (OLAM), see Methods) to assess the impact of soil structure on global climate. We hypothesized that the current pervasive and potentially biased representations of soil hydraulic properties can have important consequences for predicting water fluxes from the ecosystem and global climate models, in terms of groundwater recharge, surface water, energy, and biogeochemical fluxes, and thus ultimately for climate.

Results at the local scale confirm the hypothesis. Including soil structure significantly modifies infiltration-runoff partitioning and recharge in wet and vegetated regions where more infiltration and less runoff occur, affecting deep drainage. However, differences in climatic variables between simulations at the global scale with and without the presence of soil structure are statistically insignificant. We discuss how this outcome is a likely consequence of the coarse spatial resolution adopted in global-scale analyses. Coarse spatial resolution cascades on the representation of physical processes, as short-intense rainfall events or lateral water redistribution, dampening the impact of soil structure effects.

## Results and discussion

**Current soil hydraulic parameterization is inherently biased.** Soil hydraulic properties are presently deduced from PTFs that were derived (trained) based on limited soil information from samples that are often clustered in a few geographic regions and obtained primarily from agricultural soils (e.g., see refs. [41–44]). This induces a bias in the inferred soil hydraulic properties for natural landscapes (especially forests) used in global land-surface models. The bias stems from two interlinked sources: the legacy of sampling primarily agricultural and arable lands, and the systematic avoidance of locations with significant roots and large voids for methodological reasons. Consequently, soil structural features related to biological activities (encompassing effects of aggregates, biopores, and macropores) that are expected to affect the hydrology and surface fluxes in many natural soils remains systematically underrepresented. From a hydrologic point of view, the primary effect of biotic soil structure is expected to manifest at and near saturated conditions, when large pores are water-filled and activated, affecting the bulk soil hydraulic conductivity and infiltration rates (Supplementary Fig. 1). The macroporosity involved is typically small (often < 5% of the soil volume); hence, the impact on soil water retention is relatively minor and the main consequence of soil structural macroporosity is an increase in the saturated hydraulic conductivity[45] (relative to values based on texture alone). We present the distribution of saturated hydraulic conductivity associated with the main soil textural classes from a widely used database (UNSODA) of soil hydraulic properties[42,46,47]. Only a fraction of collected soil samples includes structural effects; thus, the limited extent of structural features is expected to bias the true distribution of soil hydraulic properties (Fig. 1a, b). Attempts to include soil organic matter as a covariate in the PTFs (e.g., see ref. [48]) only partially alleviate this bias. For instance, estimates of saturated hydraulic conductivity ($K_s$) from simple soil properties, such as bulk density and soil organic matter, offered only low predictive capabilities[40,49,50]. We have used limited available information to determine the ratio of saturated hydraulic conductivity, $K_{s,str}/K_{s,tex}$, including structural effects and soil texture only, suggesting that $K_{s,str}$ is one to three orders of magnitude larger than $K_{s,tex}$ (Fig. 1c). Pending availability of soil hydraulic functions that explicitly consider natural soil structure, which will allow training new PTFs, we propose a simple approach to systematically account for biotic effects of soil structure on soil hydraulic functions, with effects of soil structure diminishing with depth proportionally to the cumulative root depth distribution (Methods). The primary result is manifested in the unsaturated soil hydraulic conductivity function at and near saturation, a narrow but important range during which the largest infiltration fluxes may occur (Supplementary Fig. 1).

Although several abiotic factors contribute to soil structure features[51,52], including expansion and contraction of minerals, wet–dry and freeze–thaw cycles, and chemically induced aggregation, these are not explicitly considered, because their effect on hydraulic functions remains difficult to generalize and is often seasonal[53,54]. In most soils (except some arid locations), the primary agent for soil structure formation is biological activity by formation of biopores and soil aggregates[55,56]. To capture this primary effect and as a first-order approximation, we propose linking soil structure and associated hydraulic parameterization with vegetation via local Gross Primary Production (GPP) as a surrogate for biological activity (Fig. 2). A positive correlation between proxies of vegetation productivity and enhanced hydraulic conductivity is supported by a number of observations carried out in different climates[57–61]. Concurrently, a positive correlation is observed between saturated hydraulic conductivity and soil organic content[62] or macroporosity[45]. However, the

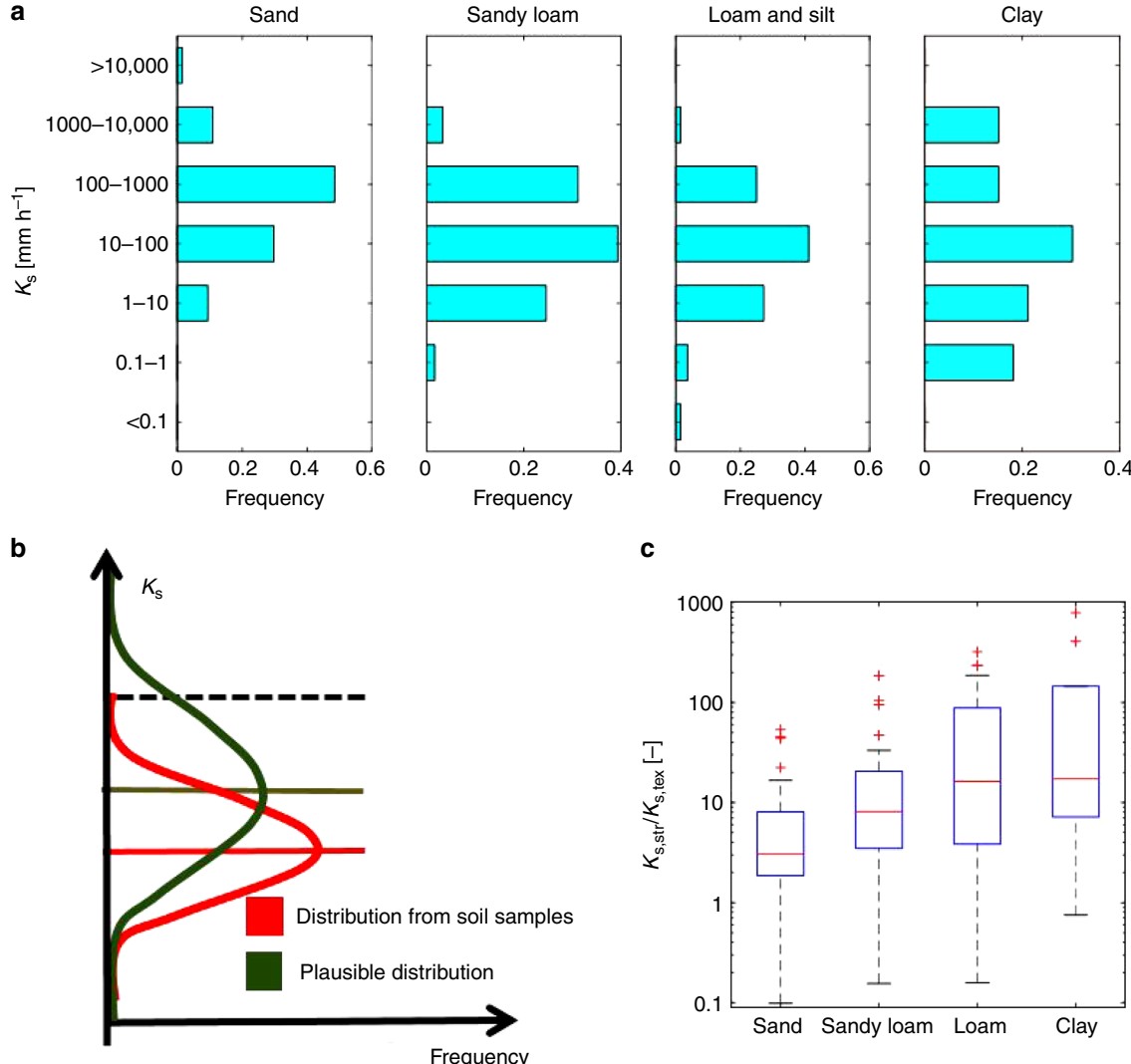

**Fig. 1 Biases in current soil hydraulic conductivity parameterizations. a** Distribution of saturated hydraulic conductivity subdivided for soil-textural types (sand, sandy loam, silt and loam combined, and clay) for undisturbed samples only of the UNSODA database (Nemes et al.[46] and Børgesen et al.[47]). **b** Hypothetical distributions of soil saturated hydraulic conductivity ($K_s$) obtainable from soil cores vs. rarely observed structured soil samples. The dashed line delimits the largest value of $K_s$ related to texture. Solid lines are the medians of the two distributions. **c** Distribution of the ratio between structural and textural saturated conductivity $K_{s,str}/K_{s,tex}$ subdivided for soil-textural types (data from Weynants et al.[39]). The box length provides the interquartile range (IQR), the bottom of the box the 25th percentile (first quartile, q1), the top of the box the 75th percentile (third quartile, q3), and the horizontal line within the box the median value. The lower whisker corresponds to q1 − 1.5IQR and the upper whisker corresponds to q3 + 1.5IQR; outliers are plotted individually.

uncertain nature of such representation and limited observability of such effects at large scales[57] necessitate a simple approach (Fig. 2, Methods, and Supplementary Fig. 2) that would support systematic evaluation of primary effects of soil structure in ESMs. The approach does not account for complex biophysical processes involved in the formation of soil aggregates, biopores, and macropores[56,63,64], which could be integrated when additional knowledge and data become available.

**Ecosystem-scale effects of soil structure.** The consequences of including first-order effects of soil structure in soil hydraulic functions (by linking soil structure with vegetation attributes) were assessed via numerical experiments performed at 20 sites spanning different climates and biomes across the globe (Supplementary Table 1). Local meteorological observations at the hourly scale for periods spanning from 3 to 31 years were used as

model inputs. Vegetation parameters were representative of local conditions (e.g., see ref. [65]). Three soil scenarios were simulated (Methods) as follows: first, soil hydraulic parameterization based on the van Genuchten model derived from a global map—the SoilGrids-250m database and the Tóth PTFs[44,66,67] without soil structural effects; second, soil hydraulic parameter derived from the same global map but with soil structural effects; and, third, original soil hydraulic parameterization based on local soil textural properties, and Saxton and Rawls PTFs[48,65].

Runoff at the surface and drainage at the bottom of the soil profile (that could represent deep soil or groundwater recharge) were significantly affected by soil structure as parameterized in the model, especially for the most productive ecosystems and for poorly drained (fine textured) soils (Fig. 3). For example, accounting for soil structure increased drainage at the soil bottom (recharge) by 1050 mm/year and decreased surface runoff by 1280 mm/year for a tropical rainforest site, which correspond

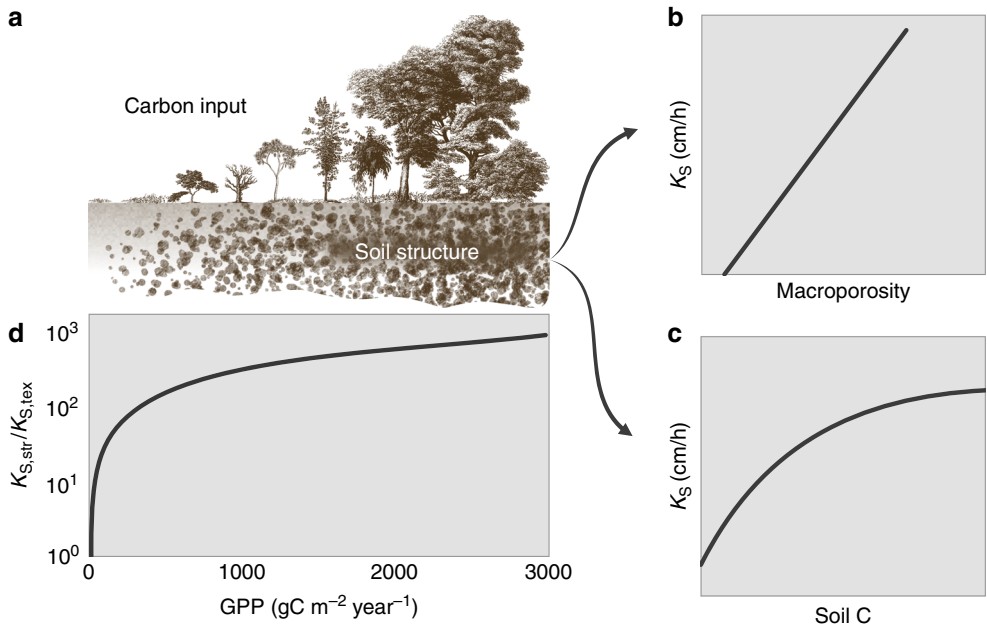

**Fig. 2 Soil structure and saturated hydraulic conductivity.** Conceptual representation of changes in saturated hydraulic conductivity induced by the presence of soil structural features (**a**). Saturated hydraulic conductivity has been shown to increase with macroporosity[45] (**b**) and with content of soil organic carbon content[62] (**c**). In this study, the ratio between structural and textural saturated conductivity $K_{s,str}/K_{s,tex}$ has been parameterized as a function of Gross Primary Production (GPP), which is used as a proxy of the degree of bioturbation activity (**d**).

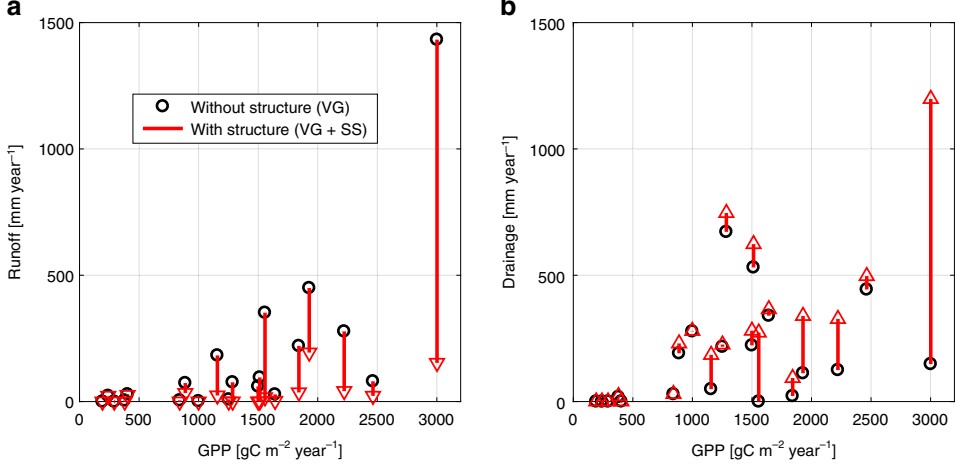

**Fig. 3 Local-scale effects in runoff and drainage.** Simulated long-term average (T&C) ecosystem-scale runoff (**a**) and water drainage at the bottom of the soil profile (**b**) for the 20 locations as a function of the local Gross Primary Production (GPP). Two scenarios are shown: VG soil hydraulic parameter (van Genuchten parameterization) derived from global maps and without soil structural effects and VG + SS soil hydraulic parameter derived from global map with soil structural effects. The red arrows indicate the magnitude of change after introducing soil structural effects.

to roughly 40–45% of annual precipitation (Manaus location). The median changes for the 20 locations of deep drainage and runoff were nearly balanced, with +46 mm/year of drainage and −48 mm/year of runoff. Even when changes were small in magnitude as in semi-arid ecosystems, they represented a considerable percentage of the water balance (e.g., for the Short Grass Steppe site, 3 mm/year difference in recharge represented a 17% change in comparison with recharge without soil structure parameterization). Comparison of simulations considering soil structure with original soil hydraulic parameterization shows considerably smaller changes in runoff and recharge (Supplementary Fig. 3). This reflects the use of local soil information and the inherent tuning of soil hydraulic properties in the T&C model aimed to avoid runoff production (significant surface runoff was not reported in any of those sites) and to reproduce ecosystem

energy and water fluxes in agreement with observations[65,68–70]. Such an adjustment of the soil hydraulic parameters derived from PTFs is quite common in hydrological modeling (e.g., see refs. [71,72]), and although it is currently practiced in many ESMs, these adjustments are poorly documented and are not systematic.

The effects of including soil structure on energy fluxes, transpiration, and vegetation metrics as GPP and Leaf Area Index (LAI) were relatively small, often <2% (when compared with soils without structural parameters), with the exception of two sites where changes reached up to 15% for latent heat (Supplementary Fig. 4). Differences were generally more pronounced (typically around 5–10%) when the simulation with soil structure (WSS) was compared with the original (site-specific) soil parameterization. This suggests that using (raw) soil textural information from global maps to derive hydraulic

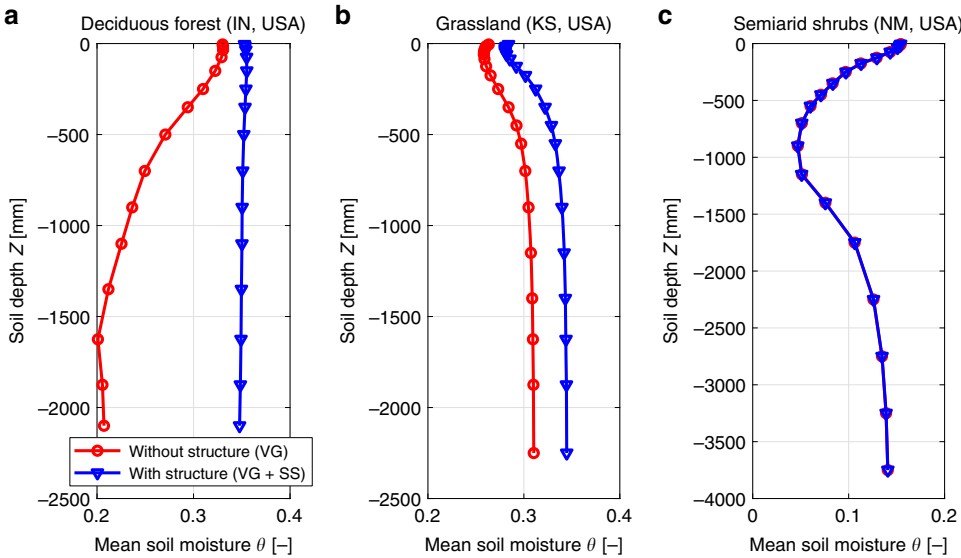

**Fig. 4 Changes in vertical profiles of soil moisture.** Long-term averaged soil moisture ($\theta$) profile with soil depth for the scenario VG and scenario VG + SS. Results correspond to simulations with T&C at the locations of Morgan Monroe Deciduous Forest (**a**), Konza Prairie Grassland (**b**), and Jornada Semi-arid shrub site (**c**).

parameters, rather than local (potentially tuned) information, may induce additional uncertainty in local terrestrial water and energy fluxes, larger even than the consideration of soil structural effects (see also ref. [31]). For some sites, soil structure modified considerably the vertical distribution of soil moisture by increasing water contents in deeper soil layers due to increased vertical redistribution under wet conditions (Fig. 4 and Supplementary Figs. 5 and 6). Soil moisture near the surface remained unaffected by soil structure (Supplementary Fig. 7), suggesting that surface evaporation and root-water-uptake dampen the effects of soil structure. The water content profile in semi-arid sites or sites characterized by well-drained sandy soils remained unchanged. This highlights the conditional manifestation of soil structure effects based on local climatic and soil textural conditions (Fig. 4).

**Global-scale effects of soil structure**. We conducted 35 years of global simulations with the soil structure representation and the original hydraulic parameterization (without structure) using the OLAM model[73–75]. We aimed to quantify large-scale effects of soil structural representation on land-surface fluxes and climate variables. The land model was first spun up for a century to establish a nearly equilibrium distribution of soil moisture and groundwater table depths. Additional 5 years of simulations were disregarded to filter out effects of arbitrary initial conditions of meteorological fields and remaining transients in soil moisture (Methods). Due to significant internal climate variability (not present in the ecosystem-scale simulations, which are run with observed climate), differences between simulations with and without soil structure were tested using a statistical two-sample $t$-test for difference in the mean of two samples. The null hypothesis was that results with and without soil structure have the same mean. Difference are analyzed separately for 11 climatic and hydrological variables, 27 geographical regions, and for the 12 months (Methods).

Differences averaged over the remaining 30 years of simulations between scenarios WSS and with no soil structure (NSS) for near-surface air temperature and vapor pressure, latent heat, and precipitation illustrate potential effects of introducing soil structure (Fig. 5). Global maps of differences in runoff and maximum daily near-surface temperature are reported in the Supplementary Fig. 8. Simulated differences in three decades of averaged temperatures were less than 0.4 °C nearly everywhere, except in certain regions such as the Canadian Arctic and central Russia, which exhibited larger differences. This result is consistent with a lack of observed changes in latent heat flux, vapor pressure, and precipitation between WSS and NSS (Fig. 5).

A systematic analysis of statistically significant differences for all analyzed cases comprising 11 variables, 27 regions, and 12 months (3564 test in total) with a significance level $\alpha = 0.05$ produced 158 cases where the means were statistically different (i.e., 4.4%); for $\alpha = 0.01$, only 43 cases or 1.2% were different. Considering that Type-1 errors (rejection of the true null hypothesis) are expected to be 5% and 1%, the result suggests that global simulations with and without soil structure parameterization resulted in statistically equivalent land-surface fluxes and climatic variables. No coherent patterns were found where one or more variables were significantly different in the two scenarios for all summer or winter months; differences appear randomly scattered (Supplementary Fig. 9). We have observed a few notable exceptions where differences (although of small magnitude) produced consistent patterns of reduced latent heat fluxes and increased air temperature in the WSS parameterization (Supplementary Fig. 10); however, the statistical significance remains weak. We emphasize that the absence of evidence for soil structure effects at the global scale does not mean that the consistent and important effects of soil structure demonstrated at small scales (Figs. 3 and 4) become irrelevant at larger scales (e.g., Fig. 5). It rather suggests that the present state-of-the-art climate model simulations at resolution of several tens of kilometers are unable to detect intermittent effects that occur on parts of the landscape and for certain rainfall intensities as we discussed further below.

**Local vs. global scale**. We introduced a first systematic attempt to modify parameterization of soil hydraulic functions induced by biotic soil structure effects, following theoretical reasoning and few-available supporting observations[39]. These new hydraulic

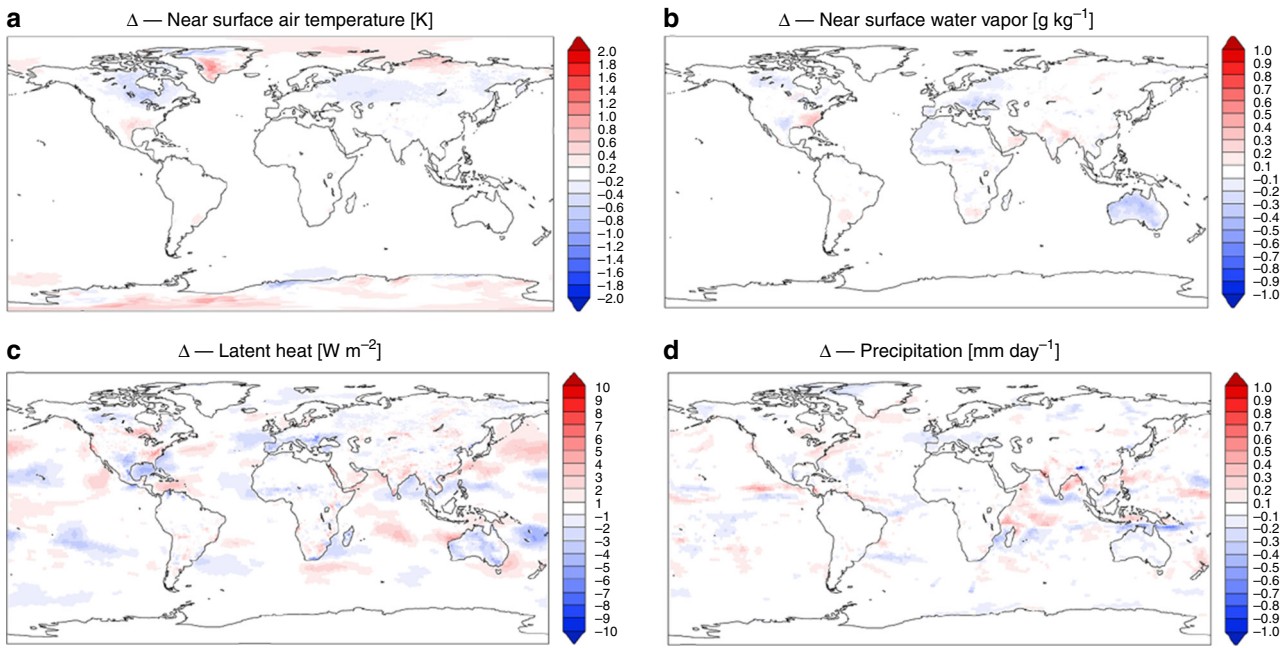

**Fig. 5 Global-scale effects on climatic variables.** Differences in (**a**) near-surface air temperature [K], (**b**) near-surface water vapor [g kg$^{-1}$], (**c**) latent heat flux [W m$^{-2}$], and (**d**) precipitation [mm day$^{-1}$] averaged over 30 years of simulations with soil structure (WSS) compared with no soil structure (NSS), i.e., $\Delta = $ WSS $-$ NSS, using the global climate model OLAM.

functions are implemented in an ecosystem model and in a global-scale climate model with a three-dimensional treatment of subsurface water fluxes. The original hypothesis that introducing soil structure generates significant differences in energy fluxes, vegetation productivity, and ultimately climate is demonstrated successfully at the ecosystem level but rejected at the global scale.

At the ecosystem scale, we have shown that including soil structural effects modify (in certain locations) the long-term hydrologic partition between relatively slow deep percolation toward groundwater table and fast surface runoff, in comparison with untuned soil parameters (Fig. 3). The vertical distribution of soil moisture is affected considerably by including structural effects in soil hydraulic functions in many locations (Fig. 4). Despite these differences, the simulated changes in energy and water fluxes are relatively minor even at these local scales (<2%). Larger changes are observed when results with soil properties derived from global maps are compared with locally tuned soil hydraulic properties (Supplementary Fig. 4), highlighting the importance of soil texture and PTF choice in affecting energy and water fluxes. Our working hypothesis that a systematic account of soil structure parameterization would propagate spatially, potentially modifying regional water resources, energy fluxes, and climate patterns, is currently unsupported by the global-scale simulations (Fig. 5). We found no consistent global-scale effects of introducing soil structure, leaving us with identification of only a number of cases, of the same magnitude of expected Type-I errors, where soil structural effects produce a statistically significant difference in a given variable and specific month (Supplementary Figs. 9 and 10).

**Spatial resolution of land and atmosphere: the elephant in the room.** The lack of global-scale evidence of soil structure effects on climatic variables may be attributed to two potential factors: the modified soil hydraulic parameterization (due to structure) is unimportant at the global scale and/or current state-of-the-art global climate models are unable to translate such small-scale soil-hydrologic processes into observable large-scale responses. Given the clear effects of soil structure at the ecosystem scale, we argue that unresolved processes at the large scale are probably responsible for the absence of effects. Theoretical considerations, small-scale soil experiments, and ecosystem-scale simulations all suggest that soil structural effects are important for water flux partitioning and for vertical water redistribution. However, the limited impact on energy fluxes even at the ecosystem scale suggests that to observe such effects requires proper accounting of lateral water fluxes for wet conditions near the soil surface. Such conditions are activated under extreme rainfall events that may last a few hours per year in some locations. The current model resolution of global simulations, e.g., ~50 km for the land surface and ~200 km for the atmosphere, is too coarse to activate such fine-scale soil structure effects. Various ongoing efforts are aimed at improving the resolution and representation of hydrological processes in coupled atmospheric-hydrological models[76,77]. However, the resolution used in most studies rarely captures details of stream networks, lateral water redistribution, saturated and unsaturated areas, and local groundwater upwelling dynamics or runoff generation. The relatively coarse resolution of global climate models results in an increase in the extent of groundwater exfiltration in arid and semi-arid regions, as well as the area of shallow water tables in northern latitudes and in tropical forests (Supplementary Fig. 11). Furthermore, the coarse resolution of the atmospheric forcing smears out high rainfall intensities, especially heavy convective events, thus reducing or completely eliminating soil structure hydrological activation. Consequently, even with soil-structural parameterization included, these features are seldom triggered in the global-scale simulations. An interesting analog is represented by flash floods, which are never generated in these global-scale simulations, but are observed in reality with evident societal implications. In retrospect, a more incremental approach would have been to use an intermediate scale (e.g., the catchment) where lateral water redistribution, saturated and unsaturated areas are better represented to see whether the spatial propagation of

soil-structural effects dampen or enhance the differences detected at the ecosystem scale. We expect that a finer spatial resolution approach would greatly improve the representation of local topography such as river valleys and their impact on the local water table and groundwater flow. Such refinement will provide more realistic conditions for assessing effects of soil structure, in particular, where the partitioning between runoff and groundwater recharge affects land-surface fluxes, as demonstrated in studies using finer spatial resolutions[23,78]. The rapid improvements in quality and resolution of soil property maps[66,67] and the development of methods capable of inversely estimating soil hydraulic parameters over the landscape based on surface temperature or streamflow[79,80] provide an impetus for injecting this forgotten aspect of soil structure in land-surface parameterization, irrespective of the lack of evidence of soil structure effects at the global scale.

**Implications for soil parameterization of ESMs.** In practice, land-surface and ESMs could be (are) tuned to implicitly account for effects of soil structure (e.g., avoiding runoff production for well-drained locations as shown by the original T&C simulations (Supplementary Fig. 3)). However, such ad-hoc tuning has not been formalized or translated into a clear framework or workflow. This ambiguity leaves a large degree of subjectivity in modifying soil hydraulic properties derived from PTFs, as well as in establishing an empirical adjustment of soil parameters, thus hindering updates as new soil information becomes available for concerns related to model performance[81]. Here we introduced a simple but systematic approach for modifying soil hydraulic parameterization that accounts for biotic soil structural features, with abiotic effects that can be added in the future provided additional knowledge is available on their impact on hydraulic functions. More generally, we expect the current method to be refined using textural dependencies and pedological information such as soil classes, parent material, qualitative soil structural descriptions[82], and, most important, availability of new quantitative data on soil structure and its relations to bioturbation. This link can be derived either empirically (e.g., see ref. [83]) or using mechanistic models[34,84,85]. The role of soil structure may be extended beyond hydraulic properties and affect modeling of soil biogeochemistry with potentially much larger global-scale implications for the carbon cycle given the modification of soil-moisture profiles[86]. We argue that future analysis with higher-spatial resolutions will allow for a more detailed representation of lateral hydrological fluxes and will thus shed further light on the effect of soil structure on ESMs.

## Methods

**Soil hydraulic functions.** Predictive functions that make use of ready available soil information to define the hydraulic properties of the soils are required. The main soil hydraulic functions are the ones relating unsaturated hydraulic conductivity $K$ [mm h$^{-1}$] with soil water content $\theta$ [−] or [mm$^3$ mm$^{-3}$], and the soil water potential $h$ [m] or [MPa] with water content $\theta$ [−], known as the soil water retention curve. The parameters of these relationships are broadly characterized with PTFs. PTFs are based on various statistical or machine-learning methods linking soil textural and physical properties to soil hydraulic parameters (e.g., see refs. [40,42–44,87,88]). Different parameterizations have been proposed in the literature to describe the hydraulic conductivity function $K = f(\theta)$ and soil water retention curve $h = f(\theta)$ (e.g., see refs. [29,89]). Arguably the van Genuchten-Mualem (VG) model [30,90] and corresponding parameterization is one of the most used approaches to describe soil hydraulic properties in hydrological and land-surface models, and is also used in this study as a basic parameterization to introduce modifications to soil hydraulic parameters induced by soil structure.

The parameters required to fully define the characteristic soil hydraulic functions are the saturated water content $\theta_{sat}$ [−], the residual water content $\theta_r$ [−], the saturated hydraulic conductivity $K_{s,tex}$ [mm h$^{-1}$], and parameters characterizing the shape of the soil water retention curve as the inverse of the air-entry pressure $\alpha_{tex}$ [mm$^{-1}$] and a parameter related to the pore-size distribution $n_{tex}$ [−], where the "tex" subscript is used to highlight that often (but not always)

these parameters are solely indicative of the soil texture (the matric soil) and not of the presence of soil structural features. According to this model, the soil water retention curve is:

$$h = \left(\frac{1}{\alpha_{tex}}\right)\left(S_e^{\left(-\frac{1}{m_{tex}}\right)} - 1\right)^{\frac{1}{n_{tex}}}, \qquad (1)$$

and the hydraulic conductivity curve is defined as:

$$K = K_{s,tex} \frac{\left(1 - |\alpha_{tex}h|^{n_{tex}-1}[1 + |\alpha_{tex}h|^{n_{tex}}]^{-m_{tex}}\right)^2}{[1 + |\alpha_{tex}h|^{n_{tex}}]^{l \cdot m_{tex}}}, \qquad (2)$$

where $l = 0.5$ is a parameter accounting for the tortuosity of the flow path, $m_{tex} = 1 - 1/n_{tex}$ and $S_e$ is the effective saturation of the soil for a given water content $\theta$:

$$S_e = \frac{\theta - \theta_r}{\theta_{sat} - \theta_r}. \qquad (3)$$

**Soil hydraulic functions in presence of soil-structural features.** Soil structure can influence the soil hydraulic functions creating a bimodal or multimodal pore-size distribution (e.g., see ref. [34]). Various soil hydraulic functions can be fitted to these pore-size distributions, including very simple models based on composite equations. In the simplest case, the porous medium can be subdivided into two regions and for each region a VG-type function can be used to describe the soil hydraulic properties (e.g., see refs. [92,93]). A linear superposition of two VG functions provides one way of describing the role of soil structure in the hydraulic properties. Adapting from Othmer et al.[92], we can write the composite soil water retention curve as:

$$\theta = \frac{\theta_{mac}}{(1 + |\alpha_{str}h|^{n_{str}})^{m_{str}}} + \frac{\theta_{sat} - \theta_{mac} - \theta_r}{(1 + |\alpha_{tex}h|^{n_{tex}})^{m_{tex}}} + \theta_r \qquad (4)$$

where $\theta_{mac}$ [−] is a macroporosity term corresponding to the saturated water content associated with the presence of structural features and is typically less than 5–10% of the total volume and $\alpha_{str}$ [mm$^{-1}$], $n_{str}$ [−], $m_{str} = 1 - 1/n_{str}$ are parameters defining how soil structure affects the shape of the soil water retention and hydraulic conductivity curves. The term $\theta_{sat}$ is the saturated water content of the bulk soil sample and can be higher than only the matric soil, which saturated water content is now $\theta_{sat} - \theta_{mac}$. The hydraulic conductivity function $K$ assuming continuity and superposition is:

$$K = K_m + K_p \qquad (5)$$

where $K_m$ repeats Eq. (2) for the textural domain and $K_P$ represents the hydraulic conductivity for the structural component:

$$K_m = K_{s,tex} \frac{\left(1 - |\alpha_{tex}h|^{n_{tex}-1}[1 + |\alpha_{tex}h|^{n_{tex}}]^{-m_{tex}}\right)^2}{[1 + |\alpha_{tex}h|^{n_{tex}}]^{0.5m_{tex}}} \qquad (6)$$

$$K_p = K_{s,str} \frac{\left(1 - |\alpha_{str}h|^{n_{str}-1}[1 + |\alpha_{str}h|^{n_{str}}]^{-m_{str}}\right)^2}{[1 + |\alpha_{str}h|^{n_{str}}]^{0.5m_{str}}} \qquad (7)$$

The term $K_{s,str}$ [mm h$^{-1}$] is the saturated hydraulic conductivity due solely to the presence of soil structure and $K_{s,tex}$ equals the saturated hydraulic conductivity determined by soil texture. It is noteworthy that using a superposition of VG functions also for the hydraulic conductivity represents an approximation, as following the Mualem derivation $K$ should have been derived integrating the effective saturation $S_e$ that include structural features (Eq. 4). However, the difference would be minimal and the Mualem integration of the bimodal conductivity function cannot be expressed in a closed form and would need to be pre-computed numerically with interpolation necessary between prescribed values, which is typically less computationally efficient than an analytical form[93]. An example of the hydraulic conductivity and soil water retention functions separated for textural and structural components is presented in Supplementary Fig. 1. The linear superposition of two VG functions allows including soil structural features in the hydraulic parameterization but still does not separate water potentials and water contents in the two domains, for instance, allowing non-equilibrium solutions, as in the case of the dual-porosity and dual permeability approach[33]. Such a simplification allows for a much simpler and efficient numerical solution required by land-surface models. A further simplification can be made assuming that the soil structure has no effect on the soil water retention curve, which is equivalent to imposing $\theta_{mac} = 0$ in Eq. (4). In this way, the soil water retention curve reduces to the same form of the classical VG parameterization and only the new hydraulic conductivity function is replaced in the model.

**Parameterizing the soil structure.** Considering a non-deformable soil, the soil hydraulic functions including soil structural features require at most four new parameters that need to be determined: the saturated hydraulic conductivity due solely to the presence of soil structure ($K_{s,str}$), the $\alpha_{str}$ and $n_{str}$ parameters affecting the shape of the hydraulic functions, and the water content at saturation associated with the structural features ($\theta_{mac}$). Even with the simplifying assumption of $\theta_{mac} = 0$, the other three parameters need to be determined. As discussed in the main text,

there is an important bias in the collection of soil samples that, even when soil is undisturbed, leaves only few samples carrying signatures of soil structure. Values for non-textural parameters are rare, because few studies report fitted parameters for both the structural and textural parts of the soil-pore distributions (e.g., see refs. [32,35,39,91,94]). We make use of this sparse information to constrain the values of $K_{s,str}$, $\theta_{mac}$, $\alpha_{str}$, and $n_{str}$. The main assumption is that the manifestation of biotic soil structural features is linearly correlated with the degree of biological activity in the soil (root growth and turnover, microbial processes, and the presence of macrofauna) and this correlates with the Net Primary Production (NPP) and thus also with GPP. Specifically, we assume that the ratio between structural and textural saturated conductivity, $K_{s,str}/K_{s,tex} = 1$ (no structural effects) without biological activity GPP = 0 gC m$^{-2}$ per year and $K_{s,str}/K_{s,tex} = 1000$, for a very high level of GPP equal to 3000 gC m$^{-2}$ per year, a characteristic value for tropical moist forests[95]. A $K_{s,str}/K_{s,tex} = 1000$ represents among the maximum values observed in literature of the difference between structural and textural saturated conductivity[37]. Between these two values, GPP is assumed to scale linearly (Supplementary Fig. 2). It is noteworthy that this is conceived as the simplest way to account for soil-structural effects, because $K_{s,str}/K_{s,tex}$ is assumed to be independent of land-use and soil texture itself, whereas there is evidence that structural features may affect a clay or a sandy soil differently[39], given the same degree of biological activity. However, we do not have enough information to parameterize the $K_{s,str}/K_{s,tex}$ dependence on soil textural classes and we intentionally keep the addition of soil structural effects as simple as possible, as this is the first attempt to include the role of soil structure in land-surface models.

Once the $K_{s,str}/K_{s,tex}$ has been determined based on GPP information, the $\alpha_{str}/\alpha_{tex}$ is computed as a function of $K_{s,str}/K_{s,tex}$. The nonlinear fit is obtained from the sparse available data, reporting both values (Supplementary Fig. 2) and excluding outliers a few values where $\alpha_{str}/\alpha_{tex}$ becomes unrealistically high (>100) based on observation limits and pore-size considerations[39,47]. The ratio $\alpha_{str}/\alpha_{tex}$ increases with increasing biological activity at low $K_{s,str}/K_{s,tex}$ values (<40), but the fitting line tends to become constant around a value of ≈30 for larger $K_{s,str}/K_{s,tex}$. Regardless, the choice of the exact $\alpha_{str}/\alpha_{tex}$ value is not critical, because the sensitivity of the soil moisture profile to the parameterization of $\alpha_{str}/\alpha_{tex}$ has been shown to be quite small and any value of $\alpha_{str}/\alpha_{tex}$ between 10 and 100 generate similar results (Supplementary Fig. 6). We are unaware of information on the $n_{str}$ parameter; for this reason, its value is assumed constant and equal to 3, which is on the high tail of the distribution of observed $n_{tex}$ and corresponds to the value of easily drainable porous media as coarse sand. This value is assumed to describe the shape of the hydraulic conductivity in the soil-structural region. Analogous to $K_{s,str}/K_{s,tex}$, the maximum water content associated with the presence of structural features $\theta_{mac}$ is assumed to be a function of GPP and to scale from 0 to 0.05, which is a relatively high value of porosity induced by macropores at sites with high vegetation productivity. The use of a larger value for maximum $\theta_{mac}$ (e.g., 0.10) do not appreciably modify the results (Supplementary Fig. 6), whereas difference could be seen for extremely high values of maximum $\theta_{mac} = 0.20$. Furthermore, change in hydraulic properties are scaled up with the distribution of root with depths; in other words, the $K_{s,str}/K_{s,tex}$ correction corresponds to the maximum potential change for a given GPP at the soil surface but becomes equal to 1 (no correction) below the rooting depth. Between land surface and rooting depth, $K_{s,str}/K_{s,tex}$ scales linearly with the cumulative fine root biomass. In this way, we assume that effects of soil structure are maximized at the soil surface but diminish to null below the rooting depth. This assumption follows the hypothesis that soil structure effects are mostly of biotic rather than abiotic origin.

**The T&C Model.** For ecosystem-scale simulations, we used the state-of-the-art mechanistic ecosystem model T&C, which simulates the main components of the hydrological and carbon cycle. T&C resolves the mass and energy budgets at the land surface and describes physiological vegetation processes including photosynthesis, phenology, carbon allocation, and tissue turnover. A detailed model description is provided in previous studies and the model has been extensively tested in several sites worldwide, including the 20 selected locations for this analysis (e.g., see refs. [68–70]). The soil column is discretized using a number of vertical layers, with increasing depth from near land surface to the bedrock. Heterogeneity in the soil hydraulic and thermal properties can be accounted for in the vertical direction. Fine root biomass is distributed vertically using various profile shapes (e.g., exponential) to describe the fine root distribution in the vertical dimension. In terms of soil hydraulic parameterizations, T&C has various options and can use either the VG or the Saxton and Rawls[48] parameterizations, which is the default mode.

**The OLAM model.** We used the OLAM model[73–75] to simulate the impact of soil structure on global-scale land-surface and climatic variables. OLAM is a global non-hydrostatic weather and climate simulation model with prescribed sea surface temperature that is partly based on the Regional Atmospheric Modeling System (RAMS[96]), which incorporates a more accurate formulation of the governing dynamics equations. OLAM can represent selected geographic areas of interest at very high resolution through the use of a variable-resolution unstructured hexagonal grid, thus retaining interactions between local and global scales that are absent with limited area models. A unique feature of OLAM's computational mesh is that it does not employ a terrain-following transformation but instead uses

horizontal grid levels that intersect topography. This improves accuracy, especially where topography is steep. Atmospheric liquid and ice processes and interactions with aerosols are represented by a microphysics parameterization[96]. Radiative properties of hydrometeors are based in part on the parameterization and interface to the Rapid Radiative Transfer Model, which evaluates radiative fluxes, scattering, absorption, and emission between the atmospheric, land, and surface water components of OLAM.

Land and ocean areas are represented on an unstructured hexagonal surface grid that can be of higher resolution and locally refined independently of the atmosphere grid, allowing, e.g., a particular catchment or watershed to be represented in greater detail. OLAM's Soil-Vegetation-Atmosphere Transfer (SVAT) sub-model is partly based on the LEAF sub-model in RAMS[97] and it operates on the surface grid. The SVAT represents energy and water storage, and fluxes at the land surface and includes vegetation. The characteristic elements are a vegetation canopy air layer; surface water ponding or snow cover; soil and bedrock; and groundwater, runoff, lakes, and river water storages. The various land covers exchange mass and energy with the atmosphere, subject to conservation laws, heat conduction, and three-dimensional Richard's equation in the saturated and unsaturated soil, turbulent transfer relationships in the canopy and atmosphere, a stomatal conductance formulation, and a radiative transfer scheme. Latent heat energy of freezing and melting are accounted for in all components of the SVAT. Water-retention curves and hydraulic conductivity functions are evaluated according to the van Genuchten[30] model with parameters estimated using the Weynants et al.[39] PTFs. Land cover, Normalized Vegetation Difference Index, and terrestrial elevation datasets were obtained from the USGS EROS Data Center. Soil and bedrock hydraulic properties, which control infiltration, percolation, and groundwater flow, are based on SoilGrids[66] and GLHYMPS[98] datasets. GPP data, used in estimating soil structure properties, is derived from the MODIS MOD17 dataset. In OLAM, soil structure is represented by its impact on hydraulic conductivity, with no assumed impact on water retention. Thus, it is only represented by the correction factor $K_{s,str}/K_{s,tex}$ rather than through modifications to the hydraulic functions. For this reason, an additional linear scaling factor to eliminate soil-structural effects as water content departs from saturation is necessary. This is incorporated such that as water potential decreases from zero to negative 10 cm (a water potential value at which most macropores are drained), the structure effect ($K_{s,str}/K_{s,tex}$) decreases from its given value to 1 (no effect).

**Local study sites.** Twenty locations covering different biomes and climate characteristics were selected for the ecosystem-scale analyses (Supplementary Table 1). Hourly meteorological inputs were derived from local observations. The observational period ranges from 3 to 31 years. The 20 sites correspond to flux towers, manipulation experiments, and experimental stations where meteorological and other information are available to force and test the model (see ref. [99]). Boundary conditions in terms of soil textural properties and depth were first derived from published site information. Concurrently, soil physical properties (e.g., sand, clay, and soil organic content) were derived from the global SoilGrids-250m database[67] for the latitude and longitude corresponding to the 20 sites and converted into VG hydraulic parameters by averaging soil physical properties over the first meter of soil and using Tóth et al.[44] PTFs. This second option is representative of what a regional or global-scale land-surface model application would use as soil hydraulic parameters. However, soil depth was maintained as inferred from local information (Supplementary Table 1). Ancillary information on vegetation types and parameters (e.g., root depth, photosynthetic capacity, and specific leaf area) were also used whenever available as a standard approach in the application of T&C. In fact, generally T&C does not use Plant Functional Types, but for each site adopts a vegetation parameterization that provide satisfactory results in terms of vegetation productivity, energy and water fluxes, and local phenology (e.g., see refs. [65,69,70]).

**Local simulations.** Three types of soil scenarios were simulated for the local case studies as follows: (i) a reference scenario (hereafter called "ORI"), which corresponds to the standard T&C soil hydraulic parameters derived from local soil textural information using the hydraulic functions and PTFs of Saxton and Rawls[48], and with hydraulic parameters adjusted for specific locations when necessary as a part of the tuning process; (ii) a standard scenario with VG soil hydraulic parameters derived from Tóth et al[44]. (hereafter called "VG") and with the underlying soil textural composition derived from the SoilGrids-250m database; (iii) a scenario including the role of soil structure where the VG soil hydraulic functions are modified to account for soil structural effects as described earlier (hereafter called "VG + SS"). In the third case, we specify values of $K_{s,str}$, $n_{str}$, $\alpha_{str}$, and $\theta_{mac}$, and modify soil hydraulic properties as a function of cumulative fine root biomass with depth. Presented results are based on simulations carried out accounting for changes in the hydraulic conductivity function only, leaving unmodified the soil water retention curve (e.g., $\theta_{mac} = 0$, in Eq. 4). We verified that such an assumption has negligible consequences, running simulations, where the macropore saturated water content ($\theta_{mac}$) is also modified to account for structural effects in the soil water retention curve and where the $\alpha_{str}/\alpha_{tex}$ ratio is modified to account for uncertainty in its determination (Supplementary Fig. 2). The sensitivity analysis to the value of $\theta_{mac}$ and $\alpha_{str}/\alpha_{tex}$ was carried out for the location of Morgan Monroe Forest, which is the one where soil moisture differences between VG and VG + SS are the most pronounced. Differences between the various simulations in

the soil water content profile (Supplementary Fig. 6), and energy and water fluxes (Supplementary Table 2) were negligible for change in $\alpha_{str}/\alpha_{tex}$ and becomes comparable to the difference between VG and VG + SS only when the maximum $\theta_{max}$ is equal to 20%, an unrealistically high value of macroporosity at ecosystem scale. Simulations with $\theta_{mac} > 0$ were about 17 times slower than with $\theta_{mac} = 0$, because Eq. (4) is not analytically invertible to determine the water potential as a function of water content and numerical interpolation of the pre-computed soil water retention curve is required.

The T&C model provides a wide range of outputs at the hourly time scale. Here we simply use long-term averaged values of GPP, total evapotranspiration, transpiration, water drainage from the bottom of the soil column, runoff, net radiation, latent heat, sensible heat, LAI, and time series of soil water content distributed in the vertical profile (Supplementary Fig. 5, 6).

**Global-scale simulations**. We ran two 35-year climate simulations with OLAM to investigate the impact of soil structure on regional and global climate. The modeled atmosphere was run with 200 km grid spacing, which is a typical resolution for climate model simulations. The surface grid was instead run with 50 km grid spacing over all continental areas, except for Antarctica, which was run at coarser resolution. First, soil water content and groundwater were spun up using a slightly coarser vertical resolution in the shallowest soil layers and longer time steps for a century-long simulation. In this simulation, surface variables were prescribed using multi-year averages for each month rather than being computed prognostically. Subsequently, two simulations of 35 years are run with soil structural effects disabled (NSS) and enabled (WSS). In these simulations, all variables including meteorological, land-surface, and soil water fluxes are computed prognostically in OLAM. The first 5 years of each simulation was disregarded as a further "spin-up" period to account for differences between prescribed and prognostic forcing, and was excluded from data analysis. Time averages of model outputs were taken over the remaining 30 years. Time-averaged results from scenarios without soil structural effects were subtracted from those with structural effects, yielding the difference that was obtained when soil structure was introduced ($\Delta$ = WSS-NSS; Fig. 4). Results from the pair of simulations, however, also contain effects of the natural variability of the atmosphere (internal climate variability) that, on a 30-year time scale, is supposed to be relatively small, except for precipitation. To infer whether soil structure produces a change that is statistically robust or not, we first looked at the differences in the variables between WSS and NSS for 11 main land-surface and climatic variables (surface short-wave radiation, precipitation, near-surface air temperature, canopy temperature daily range, soil temperature daily range, canopy-specific humidity, surface net radiation, surface sensible heat flux, surface latent heat flux, skin surface temperature, and canopy temperature), for 27 different geographical regions[100] (Supplementary Fig. 12) and for each month of the year. We detect statistically significant difference between WSS and NSS using a statistical two-sample $t$-test for difference in the mean of two samples. The test's null hypothesis was that results with and without the presence of soil structure have the same mean.

### Data availability
No new data were introduced in this article. Published dataset have been properly referenced. Detailed model results are available upon request from the corresponding author. The source data underlying Figs. 1a,c, 3, 4, and 5 are provided as a Source Data file.

### Code availability
The source code of OLAM-SOIL is available at http://olam-soil.org/download/. The source code of T&C is available at https://hyd.ifu.ethz.ch/forschung/models.html.

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

## Acknowledgements
S.F. acknowledges the support of the Stavros Niarchos Foundation and the ETH Zurich Foundation (grant ETH-29 14-2). The PIs of the FLUXNET community, the manipulation experiments, and the long-term research sites that acquired and shared data used to run T&C model simulations are deeply acknowledged.

## Author contributions
S.F., D.O. and R.W. conceived the research idea and designed the study. S.F. and R.W. ran the simulations and did the analysis together with D.O. S.F. and D.O. led the writing of the manuscript. H.V., M.H.Y., T.G., T.H., S.K., N.A. and R.A. contributed to manuscript development and revisions.

## Competing interests
The authors declare no competing interests.
