## [Peer Review File · Nature Communications]

Reviewers' comments:

Reviewer #1 (Remarks to the Author):

Review of "Soil structure – an important omission in Earth System Models"

In this paper, Fatichi et al make the case that soil structural effects are an important missing component in a variety of models. They do this through literature review and a pair of idealized sensitivity tests in which they incorporate a very simple structural adjustment to the soil model in both an ecosystem model and an earth system model. I believe this to be an important contribution because many such modeling communities are unaware of the likely importance of soil structure, and the uncertainty in soil hydraulic properties in more generally. I also appreciate the practical connection to GPP, although it is perhaps not the most rigorous connection, it makes for an "easy" entry point to ESMs, as such, this is more likely to move the field than a more theoretical approach. However, I do have some concerns that I would like to see the authors address before this can be published. In particular, I suggest adding more description of the connection to GPP, some discussion of observational constraints that could be used to evaluate the validity of incorporating soil structure in this way, using a longer averaging period in the OLAM results, and incorporating a field significance test on those results.

I look forward to seeing the authors responses,

Ethan Gutmann

National Center for Atmospheric Research

Major comments:

I would like to see more documentation of evidence (beyond intuition) that GPP correlates with soil structural effects. One can certainly make the case that both mud cracks and burrows are important in the semi-arid (low GPP) western USA. Indeed, aridity may help preserve some features such that low activity is balanced by structural longevity. While I appreciate the practical simplicity of this approach, there should be some additional justification, or there should be substantial additional caveats to the point that the results presented here merely show that there could be an impact, and not that this is a recommendation for an approach that should be used by others.

This paper would be much stronger if the authors are able to make some connection to show that these emergent effects of soil structure should be present based on observed macro-scale system

behavior (e.g. ET or runoff observations). This is an incredibly difficult challenge, and likely beyond the scope of a short communication though, but the authors hint that some of the similarities with the ORI parameter results may be because the ORI parameters were tuned to local observations. Many of the ~20 local sites, will have ET and runoff observations that could be used in comparison. At a minimum, some discussion is warranted of how such a problem might be approached (e.g. via remotely sensed data or other large scale observations.) Can they diagnose any signatures from their model's behavior (e.g. inter-variable relationships) that could be measured as a test in the future?

I'm concerned that the OLAM results are not as robust as the paper seems to suggest. For one thing, the authors note at one point that 5.9% of grid cells have a statistically significant change. Given the large number of grid points and metrics examined, would this pass a field significance test? See Wilks (2006) <https://journals.ametsoc.org/doi/10.1175/JAM2404.1>. Just to a first order, I would expect 5% of grid cells to show a 95% confidence statistically significant change purely by chance, especially given the chaotic internal variability present in the global earth system. This significance problem is compounded by the large number of fields and inter-variable relationships examined in this study.

Similarly, 10 years is too short a time period to use and expect to be able to see relatively small changes. ≥ 30 years should typically be used for a climatology, otherwise internal variability in the climate system is too big a confounding factor. I don't know if the authors are able to restart their OLAM run to include an additional 20 years of simulation for each test, or if they can reliably include 4 more years and only use the 1st year for the experimental spinup, that is not ideal, but it might give them additional statistical power in their assessment.

Minor comments:

line 24 suggest changing "biological" to "biophysical" to include non-biologic activity.

While I appreciate that the methods are separate in this publication, some documentation that the structural effects were varied with depth should be noted in the main text in the discussion section, I don't think this is present now, but perhaps I missed it.

Fig 4 and S8b, why is there so much change in, e.g., air temperature in southern Greenland and Antarctica? In a 50km model, most of this should be ice, and very low GPP ground, why do GPP connected soil changes affect it this much?

Reviewer #2 (Remarks to the Author):

This is a very interesting manuscript that examines the effect of incorporating soil structure within a global simulation model on several land-surface and climate variables. The authors argue correctly that soil structure is largely ignored in current Earth system models and that the PTFs used to estimate soil hydraulic parameters have been built on datasets that systematically avoid large macropores and coarse roots. In general, the authors lay out a convincing case for why soil structure should be incorporated into these models.

However, the conclusions of the paper are significantly hampered by the basic approach which incorporates an overly-conservative view of soil structure that restricts its effects on land-surface hydrology. The following are areas in the manuscript that illustrate this view:

1. The approach completely ignores structural effects on hydraulic properties due to expansive minerals and wet-dry cycles. (See, for instance, p3183-86.) However, expansive soils are known to be extremely important in bypass flow where, under unsaturated conditions, significant amounts of water can be channeled preferentially through dynamic macropores (e.g., van Dam, 2000) [Hydrol. Process. 14:1101-1117]. By focusing on structural effects at or near saturation (e.g., Eq. 5,7) any effects of soil structure under unsaturated conditions (especially in arid and semi-arid regions) are ignored.

2. The relationship between the $\alpha_{str}/\alpha_{tex}$ ratio and the $K_{s,str}/K_{s,tex}$ ratio discussed on p151534-539 and shown in Fig. S2c, is problematic for two reasons. (1) The variability in the data do not appear to support the fitting of any function (non-linear or otherwise) to this relationship. (Regression diagnostics should be provided in the figure to judge the appropriateness of this fit.) (2) The authors state on p151536-537 that outliers were excluded where the ratio became “unrealistically high (>100).” On what basis are values over 100 being judged to be too high? The authors do not support this assertion. When taken together (i.e., the asymptotic non-linear fit and the exclusion of high values in the regression), the α_{str} parameter is considerably restricted in this approach. This means that the air-entry potential for the structural domain is restricted from becoming too small which further translates to a significant restriction in the size of the macropores assumed in this approach.

3. The authors state on p171604-609 that an additional linear scaling factor was incorporated to eliminate the structural effect below a soil water potential of less than -10 cm. The authors don't

justify this parameter. Why would an additional factor be needed to attenuate the effects of soil structure when Eq. (7) already reduces the effect as h becomes increasingly negative? This represents an additional arbitrary restriction beyond the approach discussed in the point 1 above.

4. Although stating that macropores are typically less than 5-10% of the soil volume (p131477), the authors chose an arbitrary cutoff of 5% (p151543) for θ_{mac} which they justify by a citation that references a only a single site. A better approach would be to use the 10% value as the high point to increase the chances of observing an effect from incorporating soil structure as discussed further below.

5. By parameterizing structural variables with GPP, the approach completely ignores abiotic processes responsible for the formation of soil structure. (For example, see p41105-106 or p151516-532.) Besides shrink-swell processes discussed earlier, wet-dry cycles (even in areas without significant expansive clay minerals), freeze-thaw cycles, and chemically-induced aggregation (e.g., by carbonate), can all significantly affect the presence of structure even with low GPP. The connection to GPP in this work, therefore, only examines the influence of certain biopores and restricts the effects of soil structure.

6. A related restriction is represented by the attenuation of structural effects with depth (e.g., p161549-551) which likely underrepresents these effects especially in arid and semi-arid areas of the world (e.g., Fig. 3c). Yet, abiotic mechanisms can control the distribution of soil structure both at the surface and at depth (e.g., Southard and Buol, 1988; Vaughan et al., 2008) [SSSAJ 52:1069-1076; SSSAJ 72:660-669].

The authors argue that this approach is an evaluation of first-order effects of soil structure (p41109) without getting into complex biophysical processes. This approach would be valid if their results showed that, even with this limitation, soil structure had a clear impact on global climate. By contrast, the local hydrological implications of incorporating soil structure in the model are much more convincing in this manuscript, especially considering the conservative approach taken. Hence, with respect to global climate, the approach the authors have taken make their conclusions vulnerable to a Type II error where they may be, in essence, falsely rejecting the hypothesis that soil structure affects global climates (e.g., p71223-226). A better approach, in this case, would be test an exaggerated effect of soil structure in the model; if there is no effect in the exaggerated case, then the conclusion about the impact on global climate being elusive would be better justified and protected from this type of error. This could be done, for example, by increasing the upper limit of the θ_{mac} parameter, allowing α_{str} to greatly exceed α_{tex} as indicated by the discarded "outlier" data, and removing the scaling parameter that arbitrarily limits the structural effects below a soil water potential of -10 cm as discussed above (points 3, 4, and 6) and all without sacrificing the first-order approach. The other three points discussed above could, at the very least, be discussed to set the context for the conclusions. As it stands, the major conclusion of the

manuscript is that soil structure is important for hydrological processes such as runoff formation and water residence times. Given that this is something that is already well document in the literature, this conclusion seriously undermines the novelty of this manuscript. For this reason, I cannot recommend publication in Nature Communications without a significant reanalysis and rewrite.

In addition, I found the following minor issues in the manuscript:

p1l29 Change “deep leakage” to “deep drainage” here and throughout the manuscript including Fig. 2.

p2l52 The statement, “...quantifying the role of soil structure on soil water fluxes and transport processes remains a challenge...,” needs some justification. I recommend citing Hartemink and Minasny (2014) [Geoderma 203-231:305-317] (see section 2.4) to justify this statement.

p2l67 Missing “the” before “impact.”

p3l89 Add “the four” before “main soil textural classes.”

p3l93&98 In l93, you should direct the reader specifically to panels a and c in Fig. 1. In l98, you should direct the reader specifically to panel b. Because these are out of order with respect to the text, you should switch the places of panels b and c in Fig. 1 and update these recommended panel references in the text accordingly.

p8l261 Missing “the” after “for.”

p12l439-442 The sentence is awkward. Rerword.

p12l453 What parameter is the “intercept...of the soil water retention curve” referring to? It would normally be considered saturated water content but you’ve already listed that earlier in the sentence as a separate parameter (p12l452).

p13l476 The term, θ_{mac} , is defined here as "...the water content associated with the presence of structural features." As written, this definition is problematic and inconsistent with its use elsewhere (e.g., see the caption for Fig. S2) since nothing in this definition requires that the structural pores are completely filled by this water content. This is really a macroporosity term and should be defined as "...the saturated water content associated with the presence of structural features." Similarly, "at saturation" should be added after "features" in p14l510. Also, to what soil water potentials do the smallest macropores refer in this formulation?

p13l477 The phrase "...in absolute terms..." doesn't make sense here since percentage is, by definition, relative. I recommend rewording this to "...of the total volume."

p14l485 The phrase "...influenced by..." should be replaced with "due solely to" for accuracy.

p14l490-491 Wouldn't the Durner (1994) [WRR 32:211-223] model be an example of a closed form bimodal unsaturated hydraulic conductivity function similar to the derivation referred to in this sentence? For example, see Eq. 2.39 in the HYDRUS-1D software manual, version 4.08 (https://www.pc-progress.com/Downloads/Pgm_hydrus1D/HYDRUS1D-4.08.pdf).

p14l508 Replace "in" with "due solely to the" for accuracy and clarity.

Regarding the equations listed in the methods:

Eq. (1). The use of the absolute value operator in Eq. (2) suggests that the parameter h is assumed to represent a negative head value. However, the negative sign in front of the $(1/\alpha_{tex})$ term in Eq. (1) is not consistent with a negative head value for h (to see this, rearrange the VG equation: $S_e = [1 + (-\alpha \cdot h)^n]^{-m}$). Also, why use the symbol Ψ as soil water potential in units of MPa on p12l441 and not simply h in head units (e.g., mm or cm) which it would have to be in order for the conductivity terms to be in units of length per time (e.g., p14l485)? In order for the first part of Eq. (1) to be correct where you are equating pressure to head units (i.e., $\psi = -h$), you would have to account for the density of water and the gravitational constant. Instead of this, I recommend removing both the lowercase ψ since this is not needed and replacing the capital Ψ in the text and Fig. S1 with h for clarity.

Eq. (2). Need to replace the 0.5 in the exponent of the denominator to a parameter such as l . Placing a numeric value there masks the fact that this is really a parameter (pore connectivity) whose value is being assumed to be 0.5.

Eq. (6). This equation needlessly repeats Eq. (2). I recommend indicating in the text that K_m refers to the definition given in Eq. (2) and removing Eq. (6).

Regarding the figures:

Fig. 1a,b. What textural classes do “sand-loam” and “silt/loam” refer to? Should this be “sandy loam” and “silt loam” as used by the USDA-NRCS?

Fig. 1b,c. Switch the placements of panels b and c to match the order that they are presented in the text.

Fig. 3b. Change “KN” to “KS” in the title of that figure panel.

Fig. S2. In the caption, “...b) additional porosity...” should be changed to “...b) saturated water content...” for consistency.

Reviewer #3 (Remarks to the Author):

Dear authors, many thanks for the possibility to review your interesting paper, which indeed can be a significant contribution to the society if you improve the ms substantially. The ms reads nicely but the essentials and the better insight due to structure is indeed so weak that in its present stage it gives no better insight in the structure effects even if you like to concentrate on the global scale.

I made uncounted comments in the text, which may help to really prepare an outstanding paper, which to read is necessary.

Reply to Reviewer #1

Reviewer #1 (Remarks to the Author):

Review of "Soil structure – an important omission in Earth System Models"

In this paper, Fatichi et al make the case that soil structural effects are an important missing component in a variety of models. They do this through literature review and a pair of idealized sensitivity tests in which they incorporate a very simple structural adjustment to the soil model in both an ecosystem model and an earth system model. I believe this to be an important contribution because many such modeling communities are unaware of the likely importance of soil structure, and the uncertainty in soil hydraulic properties in more generally. I also appreciate the practical connection to GPP, although it is perhaps not the most rigorous connection, it makes for an "easy" entry point to ESMs, as such, this is more likely to move the field than a more theoretical approach. However, I do have some concerns that I would like to see the authors address before this can be published. In particular, I suggest adding more description of the connection to GPP, some discussion of observational constraints that could be used to evaluate the validity of incorporating soil structure in this way, using a longer averaging period in the OLAM results, and incorporating a field significance test on those results.

I look forward to seeing the authors responses,

Ethan Gutmann

National Center for Atmospheric Research

Reply: We thank Dr. Gutmann for the positive evaluation of the manuscript. As detailed in the responses below, the manuscript has been substantially modified; we have run 35 years of global simulations with the OLAM model and re-assessed the conclusions based on these longer simulations. We also added support for the link between vegetation productivity and changes in soil structural features (aggregation, biopores) that increase soil saturated hydraulic conductivity.

Major comments:

I would like to see more documentation of evidence (beyond intuition) that GPP correlates with soil structural effects. One can certainly make the case that both mud cracks and burrows are important in the semi-arid (low GPP) western USA. Indeed, aridity may help preserve some features such that low activity is balanced by structural longevity. While I appreciate the practical simplicity of this approach, there should be some additional justification, or there should be substantial additional caveats to the point that the results presented here merely show that there could be an impact, and not that this is a recommendation for an approach that should be used by others.

Reply: We agree that supporting the link between GPP and soil-structural effects using only theoretical arguments was unsatisfactory. We have added evidence from the literature to support this postulate using observations of saturated hydraulic conductivity and infiltration capacity across vegetation productivity gradients. For instance, Thompson et al 2010 compiled a meta-analysis of biomass-infiltration relationships from nearly 50 vegetation communities spanning a wide climatic gradient and show that infiltration capacity increased with aboveground biomass especially in water-limited ecosystems. After controlling for co-variables Niemeyer et al 2014 show that saturated hydraulic conductivity is dependent on leaf area index (a measure of vegetation productivity) and

found that forest soil had eight times more preferential flow paths than a pasture soil. Similarly, Archer et al 2013 show that field saturated hydraulic conductivity enhancement is associated with the presence of coarse roots (>2 mm diameter) creating conduits for preferential flow and a deeper organic layer in the topsoil profile under woodland hinting a connection between vegetation presence and productivity and soil structural effects. Other studies indicate a positive correlation between saturated hydraulic conductivity and soil organic content (Araya and Ghezzehei, 2019) or macroporosity (Ahuja et al 1984), both of which are associated with vegetation productivity. In the meantime, we also have conducted studies to assess structure effects on surface infiltration with different distributions of vegetation covers and soil types (Bonetti and Or, in preparation). For example, the figure R1 below is a compilation of how different vegetation attributes affect saturated hydraulic conductivity (biomass (a) and LAI (b)).

We modified the manuscript to link our theoretical arguments with this observational support (LL 111-119) and we introduced a new conceptual Figure (Fig. 2) to explain better this point.

Fig R1. Effects of vegetation biomass (a) and leaf area index (b) on saturated hydraulic conductivity.

This paper would be much stronger if the authors are able to make some connection to show that these emergent effects of soil structure should be present based on observed macro-scale system behavior (e.g. ET or runoff observations). This is an incredibly difficult challenge, and likely beyond the scope of a short communication though, but the authors hint that some of the similarities with the ORI parameter results may be because the ORI parameters were tuned to local observations. Many of the ~20 local sites, will have ET and runoff observations that could be used in comparison. At a minimum, some discussion is warranted of how such a problem might be approached (e.g. via remotely sensed data or other large scale observations.) Can they diagnose any signatures from their model's behavior (e.g. inter-variable relationships) that could be measured as a test in the future?

Reply: Demonstrating observationally at large-scale the relevance of soil-structural effects is indeed an incredibly challenging task, one that would require dedicated field experiments in soils with prominent structural features and where soil-structural features were suppressed or not allowed to develop (disruptive tillage) (in short, we are not aware of such observations at catchment or larger scales). There are ongoing efforts in this direction at smaller experimental plots (Keller et al 2017), however, data remain limited and insufficient to draw general conclusions (we mention this challenge in the revised manuscript (LL 119-122)).

The closest scale appropriate argument is related to infiltration-runoff relations. The pioneering studies of Dunne et al. 1991 and the early hydrogeography studies of L'vovich (1979) have clearly shown that with increasing vegetation cover runoff generation under high rainfall rates is significantly

reduced relative to barren soil surfaces (Fig. R2). These observations are directly linked to structural modification (and possibly other effects) induce by vegetation and stimulated biological activity in the soil. The question remains however, how this increased infiltration plays out at larger scales of interest for climate. We have shown the effects at the small scale (20 locations) and are working on developing of methodology for larger “pixel scale” (Bonetti and Or, in preparation), however, at the 50 km spatial scales it appears difficult to observe consistent signature of soil structural effects on climate. This is attributed to rarity of soil structure-activating rainfall events with intensity that exceeds infiltration capacity at sufficiently large fractions of the landscape, and the proper routing of the modified fluxes such that the effect is preserved. This would be equivalent to predicting flood footprints with a global land surface model – we know floods occur but the spatial resolution and overland flow routing are so coarse that such events would not appear in a global LSM (LL 242-265).

Fig R2. Infiltration rates as a function of rainfall rates as modified by vegetation cover (Dunne et al., 1991; Stone et al 2008).

Finally, several of the analyzed sites have latent heat observations from flux tower, but given the relative small difference in ET between the various scenarios (Fig S4) and the uncertainty in flux-tower observations, ET observations will unlikely help in placing any constraint. None of the sites do report significant surface runoff (as now stated in LL 150-152), which suggest that both the ORI soil parameterization and the “VG+SS” cases are more realistic than the standard textural-based “VG”.

I'm concerned that the OLAM results are not as robust as the paper seems to suggest. For one thing, the authors note at one point that 5.9% of grid cells have a statistically significant change. Given the large number of grid points and metrics examined, would this pass a field significance test? See Wilks (2006) <https://journals.ametsoc.org/doi/10.1175/JAM2404.1>. Just to a first order, I would expect 5% of grid cells to show a 95% confidence statistically significant change purely by chance, especially

given the chaotic internal variability present in the global earth system. This significance problem is compounded by the large number of fields and inter-variable relationships examined in this study.

Similarly, 10 years is too short a time period to use and expect to be able to see relatively small changes. ≥ 30 years should typically be used for a climatology, otherwise internal variability in the climate system is too big a confounding factor. I don't know if the authors are able to restart their OLAM run to include an additional 20 years of simulation for each test, or if they can reliably include 4 more years and only use the 1st year for the experimental spinup, that is not ideal, but it might give them additional statistical power in their assessment.

Reply: We agree with the concerns expressed in these comments - 10 years of simulations were insufficient to filter out effects of internal climate variability. We now have now extended the simulations with additional 20 years. In summary, we run simulations for 35 years, disregarding the first 5 years and analyzing the remaining 30 years. Furthermore, we have now used a proper two-sample t-test for every region, month, and variables, we analyzed. The test's null hypothesis was that results with and without soil structure have the same mean. The new results suggest that we cannot statistically detect consistent signatures of soil structure at a global-scale. Statistically significant differences with 5% and 1% levels are 4.4% and 1.2%, respectively, which are no different than the expected Type-I error. The reasons for the dichotomy between small scale and the manifestation of significant soil-structure effects at the global-scale are now discussed in the revised manuscript (LL 237-273). This absence of persistent effects of soil structure in the simulations presented, does not mean that soil structural effects could be dismissed at large scale models. Instead, we understand that surface representation at a resolution of 50 km may suppress the manifestation of specific processes (similar to the onset of flooding events), and their importance will require additional tests with more advanced models and higher resolutions. While this seems relatively obvious from this vantage point it was not anticipated when we designed the numerical experiment. The study illustrates the dangers of "linear" extrapolation of processes that dominate the small-scale responses to global scales. It also points to the need for carefully constructed large-scale models that allow quantitative assessment of processes that are expressed intermittently (i.e., hydro-climatological impacts of soil structure, flooding events, thawing processes, etc.) Critically, lack of sensitivity with present models does not imply lack of importance in the hydrological cycle.

Minor comments:

line 24 suggest changing "biological" to "biophysical" to include non-biologic activity.

Reply: Changed, even though most of the focus is indeed on biotic effects on soil structure.

While I appreciate that the methods are separate in this publication, some documentation that the structural effects were varied with depth should be noted in the main text in the discussion section, I don't think this is present now, but perhaps I missed it.

Reply: We now included a sentence (LL 104-105) to remark that effects of soil-structure are scaled up with the distribution of root with depths; in other words, the $K_{s, \text{str}}/K_{s, \text{tex}}$ correction corresponds to the maximum potential change for a given GPP at the soil surface but becomes equal to 1 (no correction) below the rooting depth.

Fig 4 and S8b, why is there so much change in, e.g., air temperature in southern Greenland and Antarctica? In a 50km model, most of this should be ice, and very low GPP ground, why do GPP connected soil changes affect it this much?

Reply: With the new simulations based on 30-year averages, changes over Greenland and Antarctica are much less significant. No effects of soil structure were included in these icy regions and we attribute these changes to teleconnections and remaining effects of internal climate variability. Climate variability is expected to be stronger in regions with low precipitation additionally, especially for Greenland, we cannot rule out certain teleconnections. For example, the Canadian arctic is a bit cooler for simulations considering soil structure effects, which might have implications for movement of air masses affecting Greenland. However, this is very speculative given the lack of statistical significant differences.

Reply to Reviewer #2

Reviewer #2 (Remarks to the Author):

This is a very interesting manuscript that examines the effect of incorporating soil structure within a global simulation model on several land-surface and climate variables. The authors argue correctly that soil structure is largely ignored in current Earth system models and that the PTFs used to estimate soil hydraulic parameters have been built on datasets that systematically avoid large macropores and coarse roots. In general, the authors lay out a convincing case for why soil structure should be incorporated into these models.

Reply: We thank the reviewer for his/her positive evaluation of the arguments we made for the consideration of soil structure in Earth system models.

However, the conclusions of the paper are significantly hampered by the basic approach which incorporates an overly-conservative view of soil structure that restricts its effects on land-surface hydrology. The following are areas in the manuscript that illustrate this view:

1. The approach completely ignores structural effects on hydraulic properties due to expansive minerals and wet-dry cycles. (See, for instance, p3183-86.) However, expansive soils are known to be extremely important in bypass flow where, under unsaturated conditions, significant amounts of water can be channeled preferentially through dynamic macropores (e.g., van Dam, 2000) [Hydrol. Process. 14:1101-1117]. By focusing on structural effects at or near saturation (e.g., Eq. 5,7) any effects of soil structure under unsaturated conditions (especially in arid and semi-arid regions) are ignored.

Reply: The reviewer is correct in the statement that the representation of soil structure is “conservative” – we opted for the simplest and most direct effects of soil structure on surface fluxes (with a particular focus on hydraulic conductivity). Furthermore, to maintain a tractable representation of soil structure effects, we linked these to vegetation, associated biological activity, and ignored abiotic processes that may be even more episodic in their influence (shrinkage cracks and freeze-thaw) and are likely to have a smaller land surface extent (coverage) than biotic effects. These simplifications and assumptions are now clarified throughout the manuscript (see new Fig. 2) together with the fact that abiotic effects could further modify hydraulic properties. In short, more structural effects have not been considered for the sake of simplicity and tractability of the analyses.

2. The relationship between the $\alpha_{str}/\alpha_{tex}$ ratio and the $K_{s,str}/K_{s,tex}$ ratio discussed on p151534-539 and shown in Fig. S2c, is problematic for two reasons. (1) The variability in the data do not appear to support the fitting of any function (non-linear or otherwise) to this relationship. (Regression diagnostics should be provided in the figure to judge the appropriateness of this fit.) (2) The authors state on p151536-537 that outliers were excluded where the ratio became “unrealistically high (>100).” On what basis are values over 100 being judged to be too high? The authors do not support this assertion. When taken together (i.e., the asymptotic non-linear fit and the exclusion of high values in the regression), the α_{str} parameter is considerably restricted in this approach. This means that the air-entry potential for the structural domain is restricted from becoming too small which further translates to a significant restriction in the size of the macropores assumed in this approach.

Reply: The reviewer would certainly appreciate the paucity of soil structure information and parameters and their effects on soil hydraulic properties. In our quest to estimate parameters for the effects of soil structure we are using present parameterization as a basis and modify these based on vegetation as surrogate for soil structure. The studies of Weynants et al 2009 and Thompson et al 2010 show, difference in the saturated hydraulic conductivity between structure and textural domains could exceed a factor of 1000. However, for the parameter alpha (the inverse of air entry value potential) as related to the largest pore size (Tuller and Or 2001) a difference of more than two orders of magnitude is not supported by literature. The distribution of the alpha parameter as obtained from the Weynants et al. and UNSODA databases (considering that some of those soil samples include soil structure) span roughly two orders of magnitude (Fig. R3).

Fig R3. Probability Density Function (PDF) and Cumulative Density Function (CDF) of the alpha parameter as obtained from the Weynants et al. and UNSODA databases.

The global distribution of the alpha parameter as reported by Montzka et al. 2017 also spans two orders of magnitude only, we thus opted to limit the structure modifications to a factor of 100. Moreover, following the suggestion of the reviewer, we tested the sensitivity of the results to the value of the alpha_str/alpha_tex correction for the location Morgan Monroe Forest location. This is the location where differences in soil moisture between the scenario “VG+SS” with soil structural effects and the scenario “VG” (without soil structural effects) are maximal (Fig. S7). As it can be seen in Fig. R4b (new Fig S6) and the new Table S2, changes in the parameterization of alpha_str/alpha_tex do not affect significantly the soil moisture profile and even less so energy and water fluxes, especially when compared to the changes simulated between VG and VG+SS scenarios. Since similar considerations can be made for the role of θ_{mac} (see replay below), impacts of soil-structural effects are mostly due to the value of $K_{s,str}/K_{s,tex}$ rather than to the other parameters, this is because soil-structure effects manifest themselves only in saturated (or nearly saturated) conditions. This has been further emphasized in the manuscript.

3. The authors state on p171604-609 that an additional linear scaling factor was incorporated to eliminate the structural effect below a soil water potential of less than -10 cm. The authors don't justify this parameter. Why would an additional factor be needed to attenuate the effects of soil structure when Eq. (7) already reduces the effect as h becomes increasingly negative? This represents an additional arbitrary restriction beyond the approach discussed in the point 1 above.

Reply: This is a practical constraint adopted for global-scale simulations only where effects of soil structure are expressed solely through changes in $K_{s, str}/K_{s, tex}$ rather than Eq. (7). As water potential goes below -10 cm, macropores are considered to be mostly empty and effects of soil structure becomes negligible. This is justified based on the definition of macropores (10 cm potential corresponding to pore larger than 150 μm at the conservative size of "macropores"). This has been clarified in the manuscript (LL 575-682).

4. Although stating that macropores are typically less than 5-10% of the soil volume (p131477), the authors chose an arbitrary cutoff of 5% (p151543) for θ_{mac} which they justify by a citation that references a only a single site. A better approach would be to use the 10% value as the high point to increase the chances of observing an effect from incorporating soil structure as discussed further below.

Reply: The basis for this lower volume is twofold: (1) to maintain a conservative estimate (not to exaggerate the volume of macroporosity), and (2) to maintain a link to root-volume, which is typically occupying less than 1% of the soil volume in the rooting zone (e.g., Gough and Seiler 2004). Therefore, the selection of 5% macroporosity was deemed to be a reasonable value of maximum macroporosity at ecosystem scale, representative of very productive ecosystems. While larger values have been definitely observed, we argue that they are unlikely representative of large areas.

Additionally, we have tested the sensitivity of the results to the maximum value assumed for θ_{mac} increasing maximum allowable θ_{mac} to 10% and 20%. We present results for the Morgan Monroe Forest location, where soil structural effects are most prominent. As seen in Fig. R4a (new Fig S6) the changes in θ_{mac} values affected near surface soil moisture, but a few cm below differences between VG and VG+SS parameterization are much more pronounced than the effects of θ_{mac} . Most important energy and water fluxes are affected by θ_{mac} only for the most extreme case (4*) (see results in the new Table S2). We now explicitly state that the assumed value of θ_{mac} can impact near surface soil moisture (LL 612-614).

Fig R4. Long-term averaged soil moisture θ , profile with soil depth for a number of scenarios. The base case (van Genuchten parameterization) without (VG) and with soil structure effects (VG+SS) are present in both subplot (a) and (b). In subplot (a) additional scenarios are presented where a macropore saturated water content θ_{mac} larger than zero is considered and then decreased and increased from the reference value of -50%, + 100% and + 200% (*0.5, *2 and *4). In subplot (b) additional scenarios are presented where soil structure effects are included and the ratio $\alpha_{str}/\alpha_{tex}$ is modified from the reference value ($\alpha_{str}/\alpha_{tex} = 33$) assuming $\alpha_{str}/\alpha_{tex} = 10$, $\alpha_{str}/\alpha_{tex} = 50$, and $\alpha_{str}/\alpha_{tex} = 100$. Results correspond to simulations with T&C at the location of Morgan Monroe Deciduous Forest

5. By parameterizing structural variables with GPP, the approach completely ignores abiotic processes responsible for the formation of soil structure. (For example, see p41105-106 or p151516-532.) Besides shrink-swell processes discussed earlier, wet-dry cycles (even in areas without significant expansive clay minerals), freeze-thaw cycles, and chemically-induced aggregation (e.g., by carbonate), can all significantly affect the presence of structure even with low GPP. The connection to GPP in this work, therefore, only examines the influence of certain biopores and restricts the effects of soil structure.

Reply: As we responded above, the approach aims to capture the most dominant structure-forming processes that are largely biological and related to plant roots and carbon inputs that feed micro and macro-fauna in soil. For certain regions (i.e., active clays in present and former river deltas), we expect abiotic soil structure effects that could add to the soil structure picture. This simplification (focusing on biotic processes only) has been clarified throughout the entire manuscript.

6. A related restriction is represented by the attenuation of structural effects with depth (e.g., p161549-551) which likely underrepresents these effects especially in arid and semi-arid areas of the world (e.g., Fig. 3c). Yet, abiotic mechanisms can control the distribution of soil structure both at the surface and at depth (e.g., Southard and Buol, 1988; Vaughan et al., 2008) [SSSAJ 52:1069-1076; SSSAJ 72:660-669].

Reply: We now stated explicitly that this representation of soil structure focused on biotic processes can underrepresent effects in semi-arid and arid regions where abiotic effects are predominant (LL 111).

The authors argue that this approach is an evaluation of first-order effects of soil structure (p41109) without getting into complex biophysical processes. This approach would be valid if their results showed that, even with this limitation, soil structure had a clear impact on global climate. By contrast, the local hydrological implications of incorporating soil structure in the model are much more convincing in this manuscript, especially considering the conservative approach taken. Hence, with respect to global climate, the approach the authors have taken make their conclusions vulnerable to a Type II error where they may be, in essence, falsely rejecting the hypothesis that soil structure affects global climates (e.g., p71223-226). A better approach, in this case, would be test an exaggerated effect of soil structure in the model; if there is no effect in the exaggerated case, then the conclusion about the impact on global climate being elusive would be better justified and protected from this type of error. This could be done, for example, by increasing the upper limit of the theta_mac parameter, allowing alpha_str to greatly exceed alpha_tex as indicated by the discarded "outlier" data, and removing the scaling parameter that arbitrarily limits the structural effects below a soil water potential of -10 cm as discussed above (points 3, 4, and 6) and all without sacrificing the first-order approach. The other three points discussed above could, at the very least, be discussed to set the context for the conclusions. As it stands, the major conclusion of the manuscript is that soil structure is important for hydrological processes such as runoff formation and water residence times. Given that this is something that is already well document in the literature, this conclusion seriously undermines the novelty of this manuscript. For this reason, I cannot recommend publication in Nature Communications without a significant reanalysis and rewrite.

Reply: We have not rejected the hypothesis that soil structure could affect global climatic processes, the novelty of this study is that, for the first time, effects of soil structure and its impact on soil hydraulic functions have been systematically implemented in an ecosystem model and in a global scale climate model for analyzing potential differences in energy fluxes, vegetation productivity and ultimately climate between the two scenarios. This effort goes well beyond the state-of-the-art of ESMs, and the results are unintuitive and new, certainly could not have been easily predicted "a priori" (LL 215-221). For instance, based on the results with T&C we would expect to see large-scale implications of including soil-structure once water is allowed to redistribute laterally. This is not because of the changes in latent and sensible heat, which are relatively small also in the ecosystem scale applications, but because the partition between surface and subsurface flow, when allowed to play a role in a distributed domain was expected to modify significantly moisture availability and thus energy partition (e.g., Maxwell and Kollet 2008). The reasons for not observing persistent soil structure effects are now discussed in the revised paper We fully agree with the reviewer that "lack of evidence" it is not "evidence of absence" sufficient for rejecting the hypothesis that soil structure has no impact globally. We are indeed very careful in not rejecting this hypothesis. However, it is clear

now that with current state-of-the-art global land-surface models, that have been expanded to solve soil processes (new soil databases, 3D Richards equation and groundwater flows) when run at “normal” resolution, lack the spatial resolution to represent soil structural effects globally. We think that this an important outcome of this study in guiding future research, design new experiments and help refine model structure.

Concerning the specific implementation of soil structure effects, see our replies and sensitivity analyses above where it is clear that the most important parameter is $K_{s, \text{str}}/K_{s, \text{tex}}$, while $\alpha_{\text{str}}/\alpha_{\text{tex}}$ and θ_{mac} have a much smaller influence on the results. In other words, we think it is reasonable to select “a priori” the most reasonable assumptions to implement soil structure given current knowledge without artificially exaggerating them.

Considering the knowledge gained by these global simulations and the discussion presented in the manuscript regarding why soil structure effects are not evident at global scale (LL 237-273), we think that other issues are more significant than the details and the inclusiveness of implementing all aspects of soil structural effects.

In addition, I found the following minor issues in the manuscript:

p1129 Change “deep leakage” to “deep drainage” here and throughout the manuscript including Fig. 2.

Reply: “deep leakage” has been substituted with “deep drainage”.

p2152 The statement, “...quantifying the role of soil structure on soil water fluxes and transport processes remains a challenge...,” needs some justification. I recommend citing Hartemink and Minasny (2014) [Geoderma 203-231:305-317] (see section 2.4) to justify this statement.

Reply: Yes, thanks for the suggestion we now added the reference to Hartemink and Minasny 2014.

p2167 Missing “the” before “impact.”

Reply: Done

p3189 Add “the four” before “main soil textural classes.”

Reply: Done

p3193&98 In 193, you should direct the reader specifically to panels a and c in Fig. 1. In 198, you should direct the reader specifically to panel b. Because these are out of order with respect to the text, you should switch the places of panels b and c in Fig. 1 and update these recommended panel references in the text accordingly.

Reply: Figure 1 has been re-drawn inverting the subplots, which are now referenced properly in the manuscript.

p81261 Missing “the” after “for.”

Reply: Corrected

p12l439-442 The sentence is awkward. Reword.

Reply: The sentence has been rephrased (LL 497-499).

p12l453 What parameter is the “intercept...of the soil water retention curve” referring to? It would normally be considered saturated water content but you’ve already listed that earlier in the sentence as a separate parameter (p12l452).

Reply: We were referring to the “ α_{tex} ” parameter, being $\Psi = 1/\alpha_{\text{tex}}$ at saturation ($S_e = 1$). But, indeed mathematically it is not exactly an intercept with the x-axis. We modified the sentence just to refer to the shape of the soil water retention curve.

p13l476 The term, theta_mac, is defined here as “...the water content associated with the presence of structural features.” As written, this definition is problematic and inconsistent with it’s use elsewhere (e.g., see the caption for Fig. S2) since nothing in this definition requires that the structural pores are completely filled by this water content. This is really a macroporosity term and should be defined as “...the saturated water content associated with the presence of structural features.” Similarly, “at saturation” should be added after “features” in p14l510. Also, to what soil water potentials do the smallest macropores refer in this formulation?

Reply: We thank the reviewer for catching this ambiguity. In the revised manuscript, we corrected the definition of θ_{mac} in the methods. θ_{mac} indeed represent the additional macroporosity associated with structural features. An example of shape of the soil water retention curve with and without soil structural effects is presented in Fig. S1, as you can see the differences are mostly concentrated between water potential of -10 cm and -100 cm.

p13l477 The phrase “...in absolute terms...” doesn’t make sense here since percentage is, by definition, relative. I recommend rewording this to “...of the total volume.”

Reply: Corrected.

p14l485 The phrase “...influenced by...” should be replaced with “due solely to” for accuracy.

Reply: Yes, corrected.

p14l490-491 Wouldn’t the Durner (1994) [WRR 32:211-223] model be an example of a closed form bimodal unsaturated hydraulic conductivity function similar to the derivation referred to in this sentence? For example, see Eq. 2.39 in the HYDRUS-1D software manual, version 4.08 (https://www.pc-progress.com/Downloads/Pgm_hydrus1D/HYDRUS1D-4.08.pdf).

Reply: In the original Durner 1994 article, the relative hydraulic conductivity function is computed by numerical evaluation of Mualem's (1976) predictive model on base of the unimodal or multimodal representation of the soil water retention curve. For the multimodal representation, the numerical evaluation is necessary because there is not an analytical form. Please note that Eq. 2.38 in the Hydrus

1D manual is equivalent to our Eq. (4) only in case $\theta_r = 0$. This could be the reason why Eq. 2.39 could be derived analytically.

p141508 Replace “in” with “due solely to the” for accuracy and clarity.

Reply: Corrected.

Regarding the equations listed in the methods:

*Eq. (1). The use of the absolute value operator in Eq. (2) suggests that the parameter h is assumed to represent a negative head value. However, the negative sign in front of the $(1/\alpha_{\text{tex}})$ term in Eq. (1) is not consistent with a negative head value for h (to see this, rearrange the VG equation: $S_e = [1 + (-\alpha * h)^n]^{-m}$). Also, why use the symbol Ψ as soil water potential in units of MPa on p121441 and not simply h in head units (e.g., mm or cm) which it would have to be in order for the conductivity terms to be in units of length per time (e.g., p141485)? In order for the first part of Eq. (1) to be correct where you are equating pressure to head units (i.e., $\psi = -h$), you would have to account for the density of water and the gravitational constant. Instead of this, I recommend removing both the lowercase ψ since this is not needed and replacing the capital Ψ in the text and Fig. S1 with h for clarity.*

Reply: We agree with the reviewer. The notation has been simplified, just using “ h ” in units of head (length).

Eq. (2). Need to replace the 0.5 in the exponent of the denominator to a parameter such as l . Placing a numeric value there masks the fact that this is really a parameter (pore connectivity) whose value is being assumed to be 0.5.

Reply: We were using this notation for simplicity. We now expressed the l parameter accounting for the tortuosity of the flow path explicitly.

Eq. (6). This equation needlessly repeats Eq. (2). I recommend indicating in the text that K_m refers to the definition given in Eq. (2) and removing Eq. (6).

Reply: Yes, it is true that Eq. (6) is merely a repetition of Eq. (2), but for sake of remarking the difference with Eq. (7) and having both of them back to back, we prefer to keep Eq. (6). However, the equivalence of Eq. (2) and (6) is now remarked.

Regarding the figures:

Fig. 1a,b. What textural classes do “sand-loam” and “silt/loam” refer to? Should this be “sandy loam” and “silt loam” as used by the USDA-NRCS?

Reply: The labels to the textural classes in Fig. 1 have been corrected. These are “Sandy Loam” and “Silt + Loam”, which have been combined.

Fig. 1b,c. Switch the placements of panels b and c to match the order that they are presented in the text.

Reply: We swapped the panels.

Fig. 3b. Change “KN” to “KS” in the title of that figure panel.

Reply: Correction

Fig. S2. In the caption, “...b) additional porosity...” should be changed to “...b) saturated water content...” for consistency.

Reply: Corrected, but we also kept “additional porosity” to be very clear.

Reply to Reviewer #3

Reviewer #3 (Remarks to the Author):

Dear authors, many thanks for the possibility to review your interesting paper, which indeed can be a significant contribution to the society if you improve the ms substantially. The ms reads nicely but the essentials and the better insight due to structure is indeed so weak that in its present stage it gives no better insight in the structure effects even if you like to concentrate on the global scale. I made uncounted comments in the text, which may help to really prepare an outstanding paper, which to read is necessary.

Reply: We thank the reviewer for his/her positive evaluation of the manuscript. As you will see, the text has been revised following your comments and to remark why even though global scale effects of soil structure representation are elusive in the current global simulations, it is still extremely important to present this type of work to a wide community and discuss its implications.

LL 25 The authors need to include the hydraulic and chemical processes affecting pore formation including the effects on pore continuity and accessibility

Reply: We thank the reviewer for these suggestions. However, the purpose is not to describe how soil structure is formed, but what are the ramifications of including soil structure on hydro-climatic response at the ecosystem and global scales. The representation of soil structural effects we provided is tightly connected to structural effects related to biotic activity and bioturbation in the soil, while we do not represent explicitly abiotic processes. This has now be clarified throughout the manuscript and in the abstract together with the fact that abiotic effects could further modify hydraulic properties (LL 108-111).

LL 30 how far can soil types which include the soil structure especially based on pedogenetic and geogenetic processes help to improve your approach.

Reply: This is an important point and indeed the relations in Fig. 1c clearly show that soil structure effects are more important in fine textured soils. This is now highlighted in the revised manuscript (LL 588-592). While additional quantitative information on soil structural effects will certainly help with refining the approach (of soil structure inclusion in land surface models), they are unlikely to modify the main conclusions of this study.

LL 43 The term texture includes in its complete meaning a complete homogenization which in itself is nowhere and never available under in situ conditions- thus, such term per se incudes a pseudo correlation and requests more detailed description to underline this weakness.

Reply: Our starting point is the standard pedotransfer functions (even the most modern ones based on the highly resolved SoilGrids), in using (and in this study modifying) this type of parametrization models tacitly make simplifications (such as using a single representative value for the global 50 km pixel size). We hope the reviewer would agree that there is little practical value in trying to delineate layering and subgrid variability unless there is data to test these refinements.

Furthermore, we remarked the fact that soil hydraulic functions based on soil textural properties only provide an inherent bias in the estimation of hydraulic properties, the most important being the saturated hydraulic conductivity (LL 63-64, 93-95).

LL 43 hifh flooding is certainly not primarily affected by texture but specific structure.

Reply: In the introduction, we simply list studies that have shown important effects of soil moisture and soil texture on various hydrological processes. Interestingly, none of these studies includes soil structure explicitly (we think that by “flooding” the reviewer means runoff generation)

LL 57 I completely agree that such approach is in itself useless because there are uncounted datasets available worldwide which contain also structure dependent hydraulic functions (like hydraulic conductivity and the retention curve data.

Reply: We are indeed making the point that water retention and hydraulic conductivity functions based primarily on soil textural information are omitting soil structural effects. However, we argue that for methodological reasons most of the existing data do not contain significant soil structural effects (methods tend to avoid inclusion of root channels, and most use uniform soil often from agricultural fields). We are aware of only a few studies (e.g., Weynants et al 2009 and see LL 574-578) that have explicitly computed effects of soil structure on saturated hydraulic conductivity (see Fig. 1) and even fewer that have proposed and tested inclusion of soil structure in large scale models (such as done in this study).

LL 63 this assumption is certainly not correct as can be easily derived from uncounted publications - the retention curve functions strongly depend on structure as well.

Reply: We have modified the text to explicitly state that the soil water retention curve is also modified by soil structural features (e.g., Fig. S1). However, as we now show through a sensitivity analysis (Fig. S6), changes in the soil moisture profile are mostly attributable to changes in saturated hydraulic conductivity rather than to structure modification of the water retention curve (e.g., through changes in α_{str} or θ_{mac}). Please see also our replies to Reviewer#2 raising a similar concern.

LL 71 This hypothesis is interesting but it excludes the effects of the climate change in itself on the pattern of these curves as drying alters the structural and proportional shrinkage ranges and impacts the model results on all scales!

Reply: Please consider that this study intends to analyze the first-order effects of soil structure (LL 113-115), which has never been done before in ecosystem models and global climate models (LL 215-221). While it is certainly, true that climate change will feedback on soil structure and will modify soil hydraulic properties, this type of implications are far beyond the scope of this article and will require a much more detailed knowledge of how climate could affect soil structural features.

LL 77 This maybe partly correct but there are certainly uncounted datasets available also for other landuse systems including those of FAO etc. I wonder how far a detailed look and comparative approach can also include such landuse effects and make the paper more complete and interesting.

Reply: As written before, we are not aware of any database (except Weynants et al 2009 and few other studies used to create Fig. S2) reporting how soil-structural effects modify soil hydraulic properties. Given the tentative nature of the proposed parameterization (see methods) of soil

structural effects, we refrain from having a parameterization, which is also dependent on soil texture and land-use as written at LL 588-592.

LL 90 for clarification it would be good if the main restrictions of the available data bases also in comparison with actually measured data for retention hydraulic conductivity curve patterns is described because your later conclusions need to also focus on such differences and uncertainty reasons.

Reply: As the reviewer knows, the effects of soil structure on the water retention curve are limited to a few percent changes in macroporosity that could slightly modify water retention (with differences vanishing at small values of water potential). In contrast, changes in hydraulic conductivity may completely alter the patterns of infiltration-runoff especially for fine textured soils under intense rainfall events. As we show in the new Fig S6 and Table S2, effects of soil structure are mostly attributable to changes in the saturated hydraulic conductivity rather than to the shape of the water retention curve (e.g., through changes in α_{str} or θ_{mac}). Therefore, in Fig. 1, which is motivating the study we prefer to focus on variability in saturated hydraulic conductivity.

LL 94 this is obvious because the analysis is already made with organic matter included in the undisturbed soil sample

Reply: Some of the studies cited did not think this was “obvious” and endeavored to systematically evaluate the effects of SOC on the performance of pedotransfer functions. Soil organic matter is definitely present in the undisturbed sample, but there is nothing obvious on the fact that including or excluding soil organic matter could change the performance of the pedotransfer functions (see also Araya and Ghezzehei 2019).

LL 98 this is correct and depends on the kind of aggregation and strength.

Reply: Ok.

LL 100 but this is at least to be proofed more precisely as uncounted data is published in international reviewed papers.

Reply: As written before, we are not aware of any database (except Weynants et al 2009 and few other studies used to create Fig. S2) reporting how soil-structural effects modify soil hydraulic properties. If there are databases or literature, which we have overlooked, we will appreciate very much to be addressed to those.

LL 105 this is at least only partly correct as the main structure formation processes are linked to swell shrink processes with enormous alterations of pore continuity and flux intensity as well as changes in isotropy while the biological processes due to earthworm activities are to be considered as add on.

Reply: The sentence has been modified to refer also to abiotic factors, which may contribute to create soil structural features (LL 108-111).

LL 106 what kind of aggregation is meant here? Please be more precise and maybe there are also uncounted papers which may help to clarify

Reply: We modified the sentence and we added two references for clarity (Passioura 1991; Oades 1993).

LL 110 however, the complex processes can be also used to underline the corresponding improvements as well as the remaining uncertainty degree if soil structure processes is rather generalized.

Reply: We agree that if additional knowledge and new data will be made available, complex processes leading to the formation of soil structure and its effect on hydraulic functions could be considered in a more detailed way. This has been remarked in the text (LL 122-124).

LL 126 can you please be more precise what is the structure input here?

Reply: We refer to soil structure as it is parameterized in the presented model. This has been clarified.

LL 142 are they even mostly incomplete? as an example what happens in Vertisols based on texture and including swell shrink processes through crack and slickensides formation or in Oxisols with huge cementation and structuring processes as well as due to pseudosand appearance.

Reply: Yes, they are largely incomplete, soil hydraulic parameters in common land-surface models and ESMs do not account for any soil structural or soil swell/shrink process. They are simply derived from textural information through PFTs or several times even from look-up table corresponding to the main soil classes (Van Looy et al 2017).

LL 157 I certainly agree if you deal with sandy soils under humid conditions (even there is the hydrophobisation effect to be included but the pseudosand effects on your modelling results includes certainly intense changes in your model inputs

Reply: As written above, please consider that this study intends to analyze the first-order effects of soil structure (LL 113-115), many of the mentioned effects are poorly described in modeling terms even at the pedon-scale and unfortunately they cannot be considered in large-scale applications yet.

LL 182 these simulations are certainly interesting but they still include an undefined degree of uncertainty by simply ignoring the effect of structure formation on the hydraulic properties both concerning the retention but also concerning the saturated /unsat. hydraulic conductivity which is not derived from van Genuchten etc.

Reply: This part of the manuscript has been considerably revised. However, please note that the global simulations do include effects of soil structure on the soil hydraulic parameterization as described in the method section.

LL 217 if you like to include this in your valuable analysis you certainly need to at least mention and discuss the tensorial properties of the hydraulic conductivity as a function of structure formation as this certainly alters the modelled results substantially (so called centennial highfloods which appear more often nowadays underline this effect as a substantial input)

Reply: We think that such degree of refinement is not present even in profile scale models such as application of Hydrus (there are simply no parameters to populate the model). Moreover, given the uncertainty in the determination of basic value of hydraulic conductivity; its proposed modification with soil structure spanning 2-3 order of magnitude (see discussion throughout the paper), and the

spatial-scales involved (100 - 1000 m for the ecosystem scale model) and 50 km for the global scale applications, we don't think that tensorial properties of the hydraulic conductivity would matter in present applications.

LL 220 but this is a consequence of the limited parameter accuracy.

Reply: These are difference between the two model simulations (with and without soil structural effects), where only hydraulic properties are modified, they are not a model to data comparison. Therefore, they are not affected by parameter accuracy, but simply by the assumptions of how we implemented soil structural effects.

LL 230 see former comments- this maybe be true but the readers would be interested to obtain more detailed coalculation maybe only for smaller regions to document the range of uncertainty.

Reply: The reason why, we approached the problem of including a parameterization of soil structural effects on hydraulic functions by using two models is exactly that we wanted to see effects at the "plot" scale (20 smaller regions, if you like) first, before moving directly to the global scale. Given the results we finally obtained, where effects of soil structure could be seen at the plot-scale but not globally, we think this approach was particularly valuable and an additional intermediate scale would have been beneficial. However, this is beyond the possibility of this study and can actually guide future research.

LL 485 the description is certainly mathematically correct and complete- however, it assumes that the capacity properties of the retention curve include the continuity effects of intensity functions like the hydraulic conductivity which is certainly at least to be discussed concerning the uncertainty.

Reply: We now clarified that we are assuming continuity of the hydraulic conductivity function. Please note that these equations are generally used to describe hydraulic conductivity in all hydrological and land-surface models and they do require continuity.

LL 510 the main assumption for the parameters is the given rigidity of the proe system which depends again on the predrying or prestrengthening of the soil system. How far are those limitations included in your valuable analysis.

Reply: We clarified that there is a basic assumption of non-deformable porous material (soil) behind the derivation of all these equations. Note that this assumption is made in the hydraulic parameterization of basically all the hydrological and land-surface models.

LL 516 A closer look to the literature would help to add valuable data for your interesting approach.

Reply: We are not aware of any additional dataset that could inform further our approach but only of this very sparse information (LL 576-579). If we have overlooked something, we would be glad to consider these additional studies.

LL 527 I agree that this is a simple approach, but why is it impossible to include textre dependent and represeative soil type specific data as they are available also world wide.

Reply: As stated above, we are not aware of many databases (except Weynants et al 2009) were effects of soil structure on saturated hydraulic conductivity are reported for multiple soils and textural

classes. In other words, there are many data to parameterize hydraulic properties for the matrix domain but very few data to infer how soil structure affects hydraulic properties.

LL 536 how far is this the consequence of your van Genuchten Mualem approach excluding the expected differences between the retention parameters and actual conductivity data.

Reply: We show in the new Fig S6 and Table S2 that the changes in the soil moisture profile are mostly attributable to changes in the saturated hydraulic conductivity rather than on the shape of the water retention curve (e.g., through changes in α_{str} or θ_{mac}). Please see also our replies to Reviewer#2 raising a similar concern.

LL 620 which ones?

Reply: The soil physical properties have been specified (LL 691).

Fig. 1 are the textural classes identical worldwide or are they adjusted (silt limit: 20, 50, 63 μ m) as transition to sand

Reply: These are the textural classes as reported in the original UNSODA database.

References

- Ahuja LR, Naney JW, Green RE, Nielsen DR (1984). Macroporosity to Characterize Spatial Variability of Hydraulic Conductivity and Effects of Land Management. *Soil Sci Soc Am J.* 48(4): 699-702,
- Araya and Ghezzehei (2019), Using machine learning for prediction of saturated hydraulic conductivity and its sensitivity to soil structural perturbations. *Water Resources Research*, 55:5715–5737.
- Archer, N.A.L., Bonell, M., Coles, N., MacDonald, A.M., Auton, C.A., and Stevenson, R. (2013). Soil characteristics and landcover relationships on soil hydraulic conductivity at a hillslope scale: A view towards local flood management. *Journal of Hydrology*, 497, pp.208-222.
- Dunne, T., W. Z. Zhang, and B. F. Aubry, (1991) Effects of rainfall, vegetation, and microtopography on infiltration and runoff, *Water Resour. Res.*, 27, 2271-2285,.
- Gough CM, Seiler JR (2004) The influence of environmental, soil carbon, root, and stand characteristics on soil CO₂ efflux in loblolly pine plantations located on the South Carolina Coastal Plain. *Forest Ecology and Management*, 191, 353– 363.
- Hartemink, A.E., Minasny, B., (2014). Towards digital soil morphometrics. *Geoderma* 230–231, 305–317
- Keller, T., Colombi, T., Ruiz, S., Manalili, M. P., Rek, J., Stadelmann, V., Wunderli, H., Breitenstein, D., Reiser, R., Oberholzer, H., Schymanski, S., Romero-Ruiz, A., Linde, N., Weisskopf, P., Walter, A., and Or, D. (2017). Long Term Soil Structure Observatory for Monitoring PostCompaction Evolution of Soil Structure, *Vadose Zone J.*, 16, <https://doi.org/10.2136/vzj2016.11.0118>,
- L'vovich, M. I. (1979). World water resources and their future. Translation from Russian by Raymond L. Nace, American Geophysical Union.
- Maxwell, R. M. & Kollet, S. J. (2008). Interdependence of groundwater dynamics and land-energy feedbacks under climate change. *Nat. Geosci.* 1, 665–669

- Montzka, C., Herbst, M., Weihermüller, L., Verhoef, A., and Vereecken, H (2017): A global data set of soil hydraulic properties and sub-grid variability of soil water retention and hydraulic conductivity curves, *Earth Syst. Sci. Data*, 9, 529-543, <https://doi.org/10.5194/essd-9-529-2017>,
- Niemeyer, R., Fremier, A., Heinse, R., Chávez, W., and DeClerck, F. (2014). Woody vegetation increases saturated hydraulic conductivity in dry tropical Nicaragua. *Vadose Zone Journal* 13(1).
- Passioura JB. (1991). Soil structure and plant growth. *Australian Journal of Soil Research* 29, 717–728.
- Stone, J.J., Paige, G.B., Hawkins, R.H., (2008). Rainfall intensity-dependent infiltration rates on rangeland rainfall simulator plots. *Transactions of the ASABE* 51 (1), 45–53
- Thompson, S.E., Harman, C.J., Heine, P., and Katul, G.G. (2010). Vegetation-infiltration relationships across climatic and soil type gradients. *Journal of Geophysical Research: Biogeosciences*, 115(G2).
- Tuller, M., and D. Or. (2001). Hydraulic conductivity of variably saturated porous media: Film and corner flow in angular pore space. *Water Resour. Res.* 37:1257–1276.
- Van Looy, K., Bouma, J., Herbst, M., Koestel, J., Minasny, B., Mishra, U., Montzka C., Nemes A., Pachepky Y.A., Padarian J., Schaap M.G., Toth B., Verhoef A., Vanderborght J., van der Ploeg M.J., Weihermüller L., Zacharias S., Zhang Y., Vereecken, H. (2017). Pedotransfer functions in Earth system science: Challenges and perspectives. *Reviews of Geophysics*, 55, 1199–1256.
- Weynants, M., Vereecken, H. & Javaux, M. (2009). Revisiting Vereecken pedotransfer functions: introducing a closed-form hydraulic model. *Vadose Zone Journal*, 8, 86–95.

Reviewers' comments:

Reviewer #1 (Remarks to the Author):

I believe the authors have provided an excellent set of revisions, and have only minor suggested comments. While the current results show little global scale effect, I think this remains a valuable contribution for a couple of reasons. First, it will become a more important effect as ESM resolutions increase, and second, the ESM community has largely overlooked the problems in current soil hydraulic property formulations, in part because there has been no globally available solution to (even theoretically) improving them. As such, the authors use of a very simple model that can be parameterized on a global scale is an important contribution.

Ethan Gutmann

Minor Comments:

Figure 1a the label 1000-10000 is missing a "1"

Given the discussion of finer-resolution ESM simulations and the importance of distributed hydrology, the authors may want to mention that systems to do this are available in the WRF-Hydro (Gochis et al 2018) and (perhaps unmaintained) pfwrf code (Maxwell et al 2010)

Gochis, D.J., M. Barlage, A. Dugger, K. FitzGerald, L. Karsten, M. McAllister, J. McCreight, J. Mills, A. RafieeiNasab, L. Read, K. Sampson, D. Yates, W. Yu, (2018). The WRF-Hydro modeling system technical description, (Version 5.0). NCAR Technical Note. 107 pages.

Maxwell, R.M., J.K. Lundquist, J.D. Mirocha, S.G. Smith, C.S. Woodward, and A.F. Tompson, 2011: Development of a Coupled Groundwater–Atmosphere Model. *Mon. Wea. Rev.*, 139, 96–116, <https://doi.org/10.1175/2010MWR3392.1>

Because these last references are work I have been involved in, I want to emphasize that there is no need to cite these; however, I want the authors to be aware of two somewhat unique approaches to improve land surface model parameters: 1) the use of remotely sensed land surface temperatures to

estimate landscape hydraulic properties (Gutmann and Small 2010), and 2) fitting pedo-transfer parameters to large scale streamflow responses (Mizukami et al 2017; and Samaniego et al 2010)

Gutmann, E.D., Small, E.E., 2010. A method for the determination of the hydraulic properties of soil from MODIS surface temperature for use in land-surface models. *Water Resources Research* 46, –n/a. doi:10.1029/2009WR008203

Mizukami, N., Clark, M.P., Newman, A.J., Wood, A.W., Gutmann, E.D., Nijssen, B., Samaniego, L., Rakovec, O., 2017. Towards seamless large-domain parameter estimation for hydrologic models. *Water Resources Research* 53, 8020–8040. doi:10.1002/2017WR020401

Samaniego, L., Kumar, R., Attinger, S., 2010. Multiscale parameter regionalization of a grid-based hydrologic model at the mesoscale. *Water Resources Research* 46, –n/a. doi:10.1029/2008WR007327

Comments towards revisions related to Ref #3:

There was a request by reviewer 3 for more discussion of how existing soil data that includes, for example, "pedogenetic and geogenetic processes" could be used. This is a great point, many soil maps include much more information than texture, yet the authors just mention in their reply that they agree that fine textured soils show a greater effect. This misses the point that pedogenesis, geologic substrate and other known quantities could be used to better inform our representation of soil structure in models. The authors are correct that there is no obvious way to include this at present, and this goes well beyond what the authors can be expected to do in the current manuscript. However, it would be great to see some mention of this for future work, how can the mapping of even just the primary soil orders (entisols, vertisols, oxisols, ...) be used? Since the focus of this paper is in part on information that is lacking from traditional soil surveys for ESM applications, it is important to include some discussion of the additional information that is present in soil surveys, and why it is not currently used in ESMs.

Reviewer #2 (Remarks to the Author):

Although the importance of soil structure in Earth System Models is somewhat diminished by the findings in the revised manuscript, the authors have made a compelling case for the need to account for soil structure in future modeling efforts and have provided a roadmap for doing so. The authors have done a good job in clarifying their findings in light of their assumptions and have simplified and streamlined their equation notation. I found their sensitivity analysis with respect to α and macroporosity in their response particularly helpful and I'm glad to see the addition of Fig. S6. I expect that this paper will make strongly needed and well-cited addition to the literature.

Reply to Reviewer #1

Reviewer #1 (Remarks to the Author):

I believe the authors have provided an excellent set of revisions, and have only minor suggested comments. While the current results show little global scale effect, I think this remains a valuable contribution for a couple of reasons. First, it will become a more important effect as ESM resolutions increase, and second, the ESM community has largely overlooked the problems in current soil hydraulic property formulations, in part because there has been no globally available solution to (even theoretically) improving them. As such, the authors use of a very simple model that can be parameterized on a global scale is an important contribution.

Ethan Gutmann

Reply: We thank Dr. Gutmann for the positive evaluation of the manuscript and encouraging words. The manuscript has been further modified following the additional comments.

Minor Comments:

Figure 1a the label 1000-10000 is missing a "1"

Reply: The label has been corrected and the figure updated.

Given the discussion of finer-resolution ESM simulations and the importance of distributed hydrology, the authors may want to mention that systems to do this are available in the WRF-Hydro (Gochis et al 2018) and (perhaps unmaintained) pfwrf code (Maxwell et al 2010)

Gochis, D.J., M. Barlage, A. Dugger, K. FitzGerald, L. Karsten, M. McAllister, J. McCreight, J. Mills, A. RafieeiNasab, L. Read, K. Sampson, D. Yates, W. Yu, (2018). The WRF-Hydro modeling system technical description, (Version 5.0). NCAR Technical Note. 107 pages.

Maxwell, R.M., J.K. Lundquist, J.D. Mirocha, S.G. Smith, C.S. Woodward, and A.F. Tompson, 2011: Development of a Coupled Groundwater–Atmosphere Model. Mon. Wea. Rev., 139, 96–116, <https://doi.org/10.1175/2010MWR3392.1>

Reply: Yes, we agree that it is important to recognize that there are ongoing efforts aimed at improving the resolution and representation of hydrological processes in coupled atmospheric-hydrological models. This is now explicitly stated and the suggested references are included (LL 252-254).

Because these last references are work I have been involved in, I want to emphasize that there is no need to cite these; however, I want the authors to be aware of two somewhat unique approaches to improve land surface model parameters: 1) the use of remotely sensed land surface temperatures to estimate landscape hydraulic properties (Gutmann and Small 2010), and 2) fitting pedo-transfer parameters to large scale streamflow responses (Mizukami et al 2017; and Samaniego et al 2010)

Gutmann, E.D., Small, E.E., 2010. A method for the determination of the hydraulic properties of soil from MODIS surface temperature for use in land-surface models. Water Resources Research 46, –n/a. doi:10.1029/2009WR008203

Mizukami, N., Clark, M.P., Newman, A.J., Wood, A.W., Gutmann, E.D., Nijssen, B., Samaniego, L., Rakovec, O., 2017. Towards seamless large-domain parameter estimation for hydrologic models. Water Resources Research 53, 8020–8040. doi:10.1002/2017WR020401

Samaniego, L., Kumar, R., Attinger, S., 2010. Multiscale parameter regionalization of a grid-based hydrologic model at the mesoscale. Water Resources Research 46, –n/a. doi:10.1029/2008WR007327

Reply: We now mentioned explicitly and referenced in the manuscript approaches capable of estimating soil hydraulic parameters directly over the landscape though inverse methods (LL 274-276).

Comments towards revisions related to Ref #3:

There was a request by reviewer 3 for more discussion of how existing soil data that includes, for example, "pedogenetic and geogenetic processes" could be used. This is a great point, many soil maps include much more information than texture, yet the authors just mention in their reply that they agree that fine textured soils show a greater effect. This misses the point that pedogenesis, geologic substrate and other known quantities could be used to better inform our representation of soil structure in models. The authors are correct that there is no obvious way to include this at present, and this goes well beyond what the authors can be expected to do in the current manuscript. However, it would be great to see some mention of this for future work, how can the mapping of even just the primary soil orders (entisols, vertisols, oxisols, ...) be used? Since the focus of this paper is in part on information that is lacking from traditional soil surveys for ESM applications, it is important to include some discussion of the additional information that is present in soil surveys, and why it is not currently used in ESMs.

Reply: We now write that the method presented in this article to include effects of soil structure can be further refined using pedological information such as soil classes, parent material, and qualitative soil structural descriptions (LL 56 and 289-292). Please however note that most of this information is categorical and thus mostly qualitative, which, currently, precludes a straightforward integration in PTFs or approaches as the one presented here.

Reply to Reviewer #2

Reviewer #2 (Remarks to the Author):

Although the importance of soil structure in Earth System Models is somewhat diminished by the findings in the revised manuscript, the authors have made a compelling case for the need to account for soil structure in future modeling efforts and have provided a roadmap for doing so. The authors have done a good job in clarifying their findings in light of their assumptions and have simplified and streamlined their equation notation. I found their sensitivity analysis with respect to alpha and macroporosity in their response particularly helpful and I'm glad to see the addition of Fig. S6. I expect that this paper will make strongly needed and well-cited addition to the literature.

Reply: We thank the reviewer for his/her positive evaluation of the manuscript and encouraging words.